# Single-cell transcriptomic profiling of the zebrafish inner ear reveals molecularly distinct hair cell and supporting cell subtypes

Tuo Shi[1,2†], Marielle O Beaulieu[3†], Lauren M Saunders[4], Peter Fabian[1], Cole Trapnell[4], Neil Segil[1,2], J Gage Crump[1]*, David W Raible[3,4,5]*

[1]Department of Stem Cell Biology and Regenerative Medicine, Keck School of Medicine, University of Southern California, Los Angeles, United States; [2]Caruso Department of Otolaryngology-Head and Neck Surgery, Keck School of Medicine, University of Southern California, Los Angeles, United States; [3]Department of Otolaryngology-Head and Neck Surgery, University of Washington, Seattle, United States; [4]Department of Genome Sciences, University of Washington, Seattle, United States; [5]Department of Biological Structure, University of Washington, Seattle, United States

*For correspondence:
gcrump@med.usc.edu (JGageC);
draible@uw.edu (DWR)

†These authors contributed equally to this work

**Abstract** A major cause of human deafness and vestibular dysfunction is permanent loss of the mechanosensory hair cells of the inner ear. In non-mammalian vertebrates such as zebrafish, regeneration of missing hair cells can occur throughout life. While a comparative approach has the potential to reveal the basis of such differential regenerative ability, the degree to which the inner ears of fish and mammals share common hair cells and supporting cell types remains unresolved. Here, we perform single-cell RNA sequencing of the zebrafish inner ear at embryonic through adult stages to catalog the diversity of hair cells and non-sensory supporting cells. We identify a putative progenitor population for hair cells and supporting cells, as well as distinct hair and supporting cell types in the maculae versus cristae. The hair cell and supporting cell types differ from those described for the lateral line system, a distributed mechanosensory organ in zebrafish in which most studies of hair cell regeneration have been conducted. In the maculae, we identify two subtypes of hair cells that share gene expression with mammalian striolar or extrastriolar hair cells. In situ hybridization reveals that these hair cell subtypes occupy distinct spatial domains within the three macular organs, the utricle, saccule, and lagena, consistent with the reported distinct electrophysiological properties of hair cells within these domains. These findings suggest that primitive specialization of spatially distinct striolar and extrastriolar hair cells likely arose in the last common ancestor of fish and mammals. The similarities of inner ear cell type composition between fish and mammals validate zebrafish as a relevant model for understanding inner ear-specific hair cell function and regeneration.

## Editor's evaluation

This important study describes transcriptomic profiles of sensory and non-sensory cells of the zebrafish inner ear at single-cell resolution in embryonic through adult stages. These solid results catalogue transcriptomic data and show evidence that distinct cell subtypes exist between cells of the ear and the lateral line as well as within subcellular compartments in the inner ear. These findings provide information towards comparative studies of inner ear hair cell function and regeneration.

**Figure 1.** Anatomy of zebrafish and mouse inner ears. (**A**) Illustration of the lateral line system of a 5 dpf zebrafish. Blue circles represent individual neuromasts located on the body of the fish. Boxed region indicates location of the ear. (**B**) Enlarged diagram of the 5 dpf zebrafish ear showing cristae (red) and macular (blue) sensory organs. (**C,D**) Illustrations of adult zebrafish and mouse inner ears showing homologous end organs in the semicircular canal crista ampullaris (red) and macula otolith organs (blue). Light green and dark green represent unique end organs of the lagena in zebrafish and cochlea in mice. (**E**) Illustration of the mouse utricle showing striolar and extrastriolar regions of the sensory organ. Arrows represent hair cell planar polarity within the sensory organ and red dashed line represents the line of polarity reversal within the striola. ac: anterior crista, c: cochlea, l: lagena, lc: lateral crista, o: otolith, pc: posterior crista, s: saccule, u: utricle.

## Introduction

Mechanosensory hair cells of the inner ear are responsible for sensing sound and head position in vertebrates. Hair cells are notoriously susceptible to damage from multiple types of insults, including noise and ototoxic drug exposure. Studies of hair cell physiology in mammals are limited by the location of the inner ear within the temporal bone, which precludes many targeted manipulations and in vivo imaging beyond the neonatal stage. As a result, non-mammalian vertebrates with analogous, more easily accessible hair cells have become useful models for studying hair cell development, death, and regeneration. Non-mammalian vertebrates such as birds and fish can regenerate hair cells of the auditory and vestibular systems that are lost due to injury (*Stone and Cotanche, 2007*; *Monroe et al., 2015*). This differs from mammals, where cochlear hair cell death leads to permanent hearing loss (*Corwin and Cotanche, 1988*; *Yamasoba and Kondo, 2006*), and limited regeneration of vestibular hair cells results in minimal recovery of function (*Golub et al., 2012*). Non-mammalian model systems of hair cell regeneration have the potential to reveal conserved pathways that can be targeted to promote hair cell survival and regeneration in humans. However, the extent of hair cell molecular homology across vertebrates remains unclear.

Due to its accessibility for manipulation and imaging, the zebrafish lateral line system has been widely used to study mechanisms of hair cell physiology (*Pickett and Raible, 2019*; *Sheets et al., 2021*). The lateral line is an external sensory system that allows aquatic vertebrates to detect local movement of water. Sensory organs of the lateral line, called neuromasts, contain hair cells and supporting cells that share properties with those of the inner ear. However, relative to the lateral line, cells in the zebrafish inner ear are likely more similar to their mammalian counterparts, raising the potential for it to be a more comparable system in which to study hair cell function.

Zebrafish and mammals share several inner ear sensory organs. Three semicircular canals with sensory end organs called cristae sense angular rotation of the head. Two additional sensory end organs detect linear acceleration and gravity: the utricular and saccular macula each with an associated otolith crystal (*Figure 1*). Fish lack a specific auditory structure such as the mammalian cochlea and instead sense sound through the saccule, utricle, and a third otolith organ, the lagena. Although

historically the utricle was thought to be for vestibular function and the saccule and lagena analogous to the cochlea for sound detection, there is now substantial evidence for all three otolith end organs being used for sound detection with diverse specializations across fishes (*Popper and Fay, 1993*). Zebrafish exhibit behavioral responses to sound frequencies between 100 and 1200 Hz (*Zeddies and Fay, 2005*; *Bhandiwad et al., 2013*), and neural responses up to 4000 Hz (*Poulsen et al., 2021*). In larval zebrafish, both saccule and utricle hair cells respond to vibration stimuli, with the utricle responding to relatively lower frequencies than the saccule, as well as additive effects when both are stimulated (*Yao et al., 2016*; *Favre-Bulle et al., 2020*).

Within the mammalian utricle and saccule, there are both morphological and spatial differences between hair cells (*Lysakowski and Goldberg, 2004*; *Eatock and Songer, 2011*). Hair cells are broadly classified by their morphology and innervation, with Type I hair cells having calyx synapses surrounding the hair cell body and Type II hair cells having bouton synapses. Both Type I and Type II cells can be found within the central region of the macular organs known as the striola and in the surrounding extrastriolar zones. Although the role of spatial segregation into striolar versus extrastriolar zones has not been fully elucidated, hair cells across these regions vary in morphology, electrophysiology, and synaptic structure (*Desai et al., 2005*; *Li et al., 2008*). The striola is characterized by hair cells with taller ciliary bundles and encompasses a line of polarity reversal where hair cells change their stereocilia orientation (*Figure 1E*). Whereas distinct Type I and Type II hair cells, and in particular the calyx synapses typical of Type I cells, have not been identified in the maculae of fishes, afferent innervation with some calyx-like properties has been reported in goldfish cristae (*Lanford and Popper, 1996*). Spatial heterogeneity in the maculae, including those of zebrafish, has also been previously noted (*Chang et al., 1992*; *Platt, 1993*; *Collin et al., 2000*; *Liu et al., 2022*). However, the homologies of cells at the cellular and molecular levels have remained unknown.

Recent single-cell and single-nucleus RNA-sequencing efforts have generated a wealth of transcriptomic data from hair cells in several model systems, facilitating more direct comparison of cell types and gene regulatory networks between species. Although single-cell transcriptomic data have recently been published for the zebrafish inner ear (*Jimenez et al., 2022*; *Qian et al., 2022*), the diversity of hair cell and supporting cell subtypes has not been thoroughly analyzed. In order to better understand the diversification of cell types in the zebrafish inner ear, and their relationships to those in mammals, here we perform single-cell and single-nucleus RNA sequencing of the zebrafish inner ear from embryonic through adult stages. We find that hair and supporting cells from the zebrafish inner ear and lateral line are transcriptionally distinct, and that hair and supporting cells differ between the cristae and maculae. All of these distinct cell types are present during larval development and are maintained into adulthood. In situ hybridization reveals that these hair cell subtypes occupy distinct spatial domains within the utricle, saccule, and lagena, and computational comparison of hair cell types reveals homology with striolar and extrastriolar hair cell types in mammals. These findings point to an origin of striolar and extrastriolar hair cell types in at least the last common ancestor of fish and mammals.

## Results

### Inner ear hair cells and supporting cells are distinct from those of the lateral line

To assess differences between inner ear and lateral line cells, we analyzed a subset of cells from a large single-nucleus RNA-seq dataset of whole zebrafish at embryonic and larval stages (36–96 hours post-fertilization (hpf)), which was prepared by single-nucleus combinatorial indexing and sequencing ('sci-Seq'; *Saunders et al., 2022*). Within an initial dataset of 1.25 million cells from 1233 embryos spanning 18 timepoints between 18 and 96 hr (see *Saunders et al., 2022* for more detail), a total of 16,517 inner ear and lateral line cells were isolated, combined, and re-processed using Monocle 3 (*Figure 2A–B*). Initially, otic vesicle and lateral line cell clusters were identified by *eya1* expression (*Sahly et al., 1999*) in combination with the following known marker genes. Inner ear nonsensory cells were identified by expression of the transcription factor gene *sox10* (*Dutton et al., 2009*) in combination with inner ear supporting cell genes (*stm*, *otog*, *otogl*, *otomp*, *tecta*, and *oc90*; *Figure 2C*; *Söllner et al., 2003*; *Kalka et al., 2019*; *Petko et al., 2008*; *Stooke-Vaughan et al., 2015*). Lateral line nonsensory cells were identified by expression of known markers *fat1b*, *tfap2a*, *tnfsf10l3*, *lef1*,

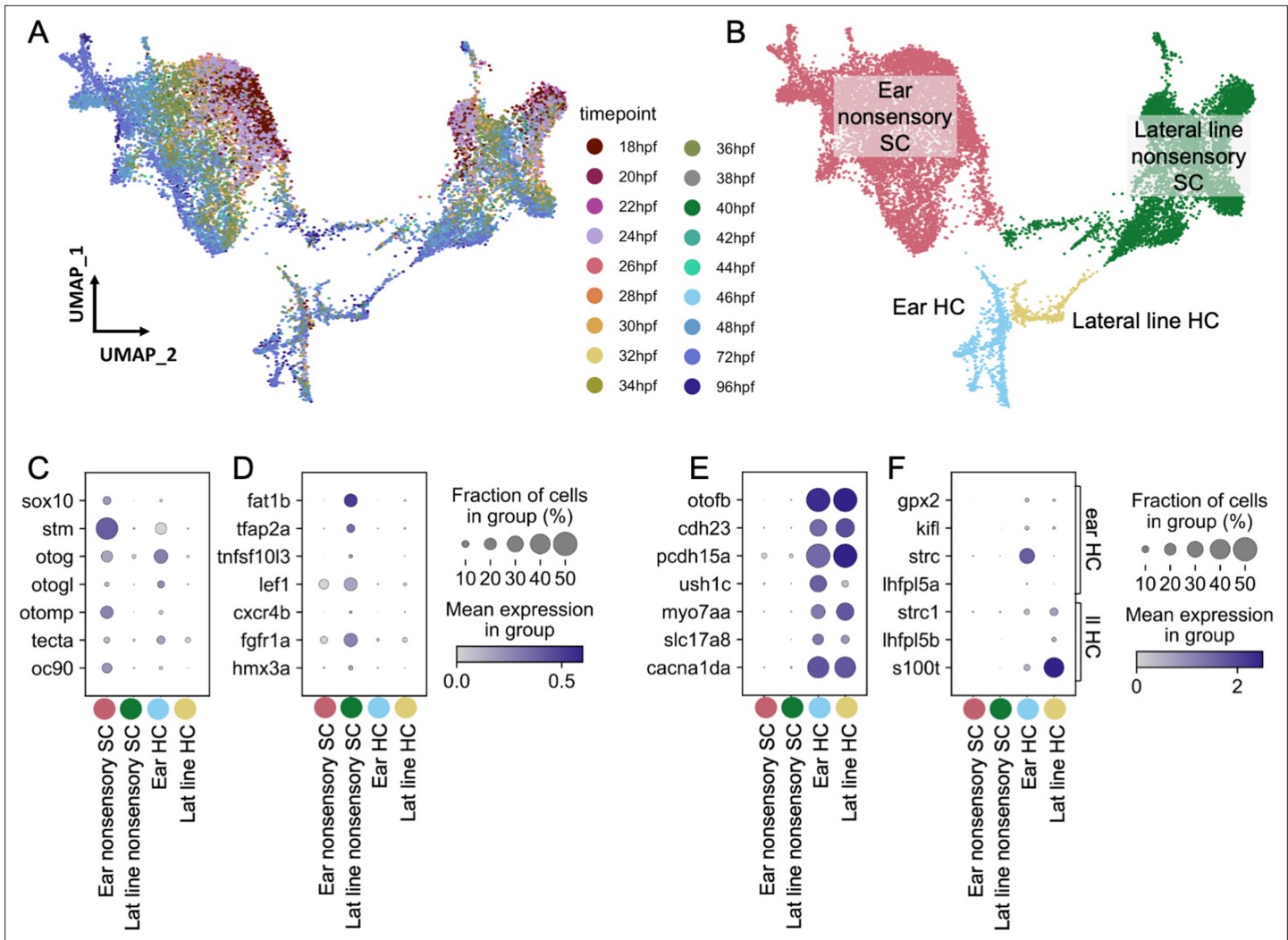

**Figure 2.** Molecularly distinct cell types between the zebrafish inner ear and lateral line. Ear and lateral line cells were selected from a whole-embryo single-nucleus RNA-seq dataset from animals between 18 and 96 hpf using known marker genes for hair cells and supporting cells. (**A–B**) UMAP projection of inner ear and lateral line cells grouped by (**A**) developmental timepoint and (**B**) broad cell type: ear nonsensory SC (red), lateral line nonsensory SC (green), ear HC (blue), and lateral line HC (yellow). Clusters in (**B**) correspond to columns of following gene expression plots. Widely accepted marker genes for (**C**) inner ear nonsensory cells, (**D**) lateral line nonsensory cells, and (**E**) hair cells show enriched expression in the corresponding clusters from B, confirming their identity. (**F**) Expression of previously identified marker genes for inner ear or lateral line hair cells was used to identify hair cell origin.

The online version of this article includes the following figure supplement(s) for figure 2:

**Figure supplement 1.** Gene modules for embryonic to larval inner ear and lateral line dataset.

**Figure supplement 2.** Selection of otic sensory cells from snRNA-seq dataset.

**Figure supplement 3.** Gene expression differences between lateral line and inner ear hair cells.

*cxcr4b, fgfr1a*, and *hmx3a* (**Figure 2D**; **Steiner et al., 2014**; **Thomas and Raible, 2019**; **McGraw et al., 2011**; **Haas and Gilmour, 2006**; **Lee et al., 2016**; **Feng and Xu, 2010**). We identified hair cells by expression of the pan-hair cell genes *otofb, cdh23, pcdh15a, ush1c, myo7aa, slc17a8*, and *cacna1da* (**Figure 2E**; **Chatterjee et al., 2015**; **Söllner et al., 2004**; **Seiler et al., 2005**; **Phillips et al., 2011**; **Ernest et al., 2000**; **Obholzer et al., 2008**; **Sheets et al., 2012**). To distinguish between inner ear and lateral line hair cells, we queried expression of previously described markers for inner ear (*gpx2, kifl, strc*, and *lhfpl5a*) and lateral line (*strc1, lhfpl5b*, and *s100t*) (**Erickson et al., 2019**; **Erickson and Nicolson, 2015**). Although many of these markers are at low abundance, these populations are marked distinctly by *strc* and *s100t* (**Figure 2F**). We used Monocle3 to identify differentially expressed

genes (*Supplementary file 1*) and to generate modules of co-expressed genes (*Figure 2—figure supplement 1*, *Supplementary file 2*).

Both hair cells and nonsensory supporting cells from the inner ear and lateral line formed distinct clusters, with nonsensory cells from the two mechanosensory organs showing greater distinction than hair cells (*Figure 2B*, *Figure 2—figure supplement 2A*). To confirm the relative differences between inner ear and lateral line hair cells and nonsensory cells, Partition-based Graph Abstraction (PAGA) analysis was used to measure the connectivity of clusters (*Wolf et al., 2019*). PAGA analysis revealed strong connectivity within inner ear supporting cell clusters and within lateral line supporting cell clusters but little connectivity between them (*Figure 2—figure supplement 2A*, *Supplementary file 3*).

The inner ear nonsensory cluster includes structural cells forming the otic capsule, identified by expression of the extracellular matrix protein-encoding genes *collagen type 2* a1a (*col2a1a*) and *matrilin 4* (*matn4*) (*Xu et al., 2018*), as well as sensory supporting cells expressing *lfng* (*Figure 3D*; *Figure 2—figure supplement 2B*). Inner ear and lateral line supporting cells remain as distinct clusters even when structural *matn4+* cells are excluded from analysis (*Figure 2—figure supplement 2C*). Thus, both hair cells and supporting cells have distinct gene expression profiles between the inner ear and lateral line at embryonic and larval stages.

## Single-cell RNA-seq reveals distinct hair cell and supporting cell populations in the juvenile and adult inner ear of zebrafish

To identify distinct subtypes of inner ear hair cells and supporting cells from larval through adult stages, we first re-analyzed single-cell RNA sequencing (scRNA-seq) datasets from larval stages (72 and 120 hpf) (*Fabian et al., 2022*), in which otic placode cells and their descendants were labeled with *Sox10*:Cre to induce recombination of an ubiquitous *ubb*:LOXP-EGFP-STOP-LOXP-mCherry transgene (*Kague et al., 2012*). We also performed additional scRNA-seq using these transgenic lines by dissecting ears from juvenile (14 days post-fertilization (dpf)), and adult (12 months post-fertilization (mpf)) animals. Following cell dissociation and fluorescence-activated cell sorting (FACS) to purify mCherry + cells, we constructed scRNA-seq libraries using 10x Chromium technology. For all datasets, hair cells and supporting cells were identified for further analysis based on the expression of hair cell markers *myo6b* and *strc* and supporting cell markers *stm* and *lfng*; structural cells were removed from further analysis based on expression of *matn4* and *col2a1a* (*Figure 3—figure supplement 1*). Using Seurat, we integrated this dataset with the sci-Seq embryonic and larval dataset (36–96 hpf) (*Figure 3A and B*). The combined dataset comprises 3246 inner ear cells separated into 10 groups based on unsupervised clustering, with differentially expressed genes for each cluster shown in *Figure 3E* and *Supplementary file 4*. We identified six clusters of hair cells based on shared expression of *myo6b*, *strc*, *lhfpl5a*, and *gfi1aa* (*Yu et al., 2020*), a nascent hair cell cluster based on expression of *atoh1a* (*Millimaki et al., 2007*) and the Notch ligand *dla* (*Riley et al., 1999*), and two clusters of supporting cells based on expression of *lfng* and *stm* (*Figure 3C and D*, *Figure 3—figure supplement 2*). An additional putative progenitor cluster (cluster 0), enriched for cells from embryonic stages, is characterized by expression of genes such as *fgfr2* (*Rohs et al., 2013*), *fat1a* (*Down et al., 2005*), *igsf3*, and *pard3bb* (*Figure 3—figure supplement 3*). Although these marker genes are differentially expressed in the putative progenitor cluster, some of them (e.g. *fat1a* and *pard3bb*) retain a lower expression level in supporting cell populations (*Figure 3—figure supplement 3F*). This is further demonstrated by gene modules of these clusters (*Figure 3—figure supplement 4*, *Supplementary file 5*), where the progenitor signature module genes (Module 1) are expressed in lower levels in the supporting cell clusters. This transcriptional relatedness between progenitors and supporting cells may underlie the role of supporting cells as a resident stem cell population during zebrafish hair cell regeneration.

## Developmental trajectories in the inner ear

To understand potential lineage relationships between clusters, we performed pseudotime trajectory analysis using Monocle3. We anchored the pseudotime projection at the putative progenitor cell cluster. Analysis revealed two major trajectories toward hair cell and supporting cell clusters for both maculae and cristae (*Figure 4A and B*, *Figure 4—figure supplement 1*), with distinct patterns of gene expression along each trajectory (*Supplementary file 6*). We find that average gene expression of the putative progenitor (Cluster 0) markers follow two patterns: decreasing along both hair cell and

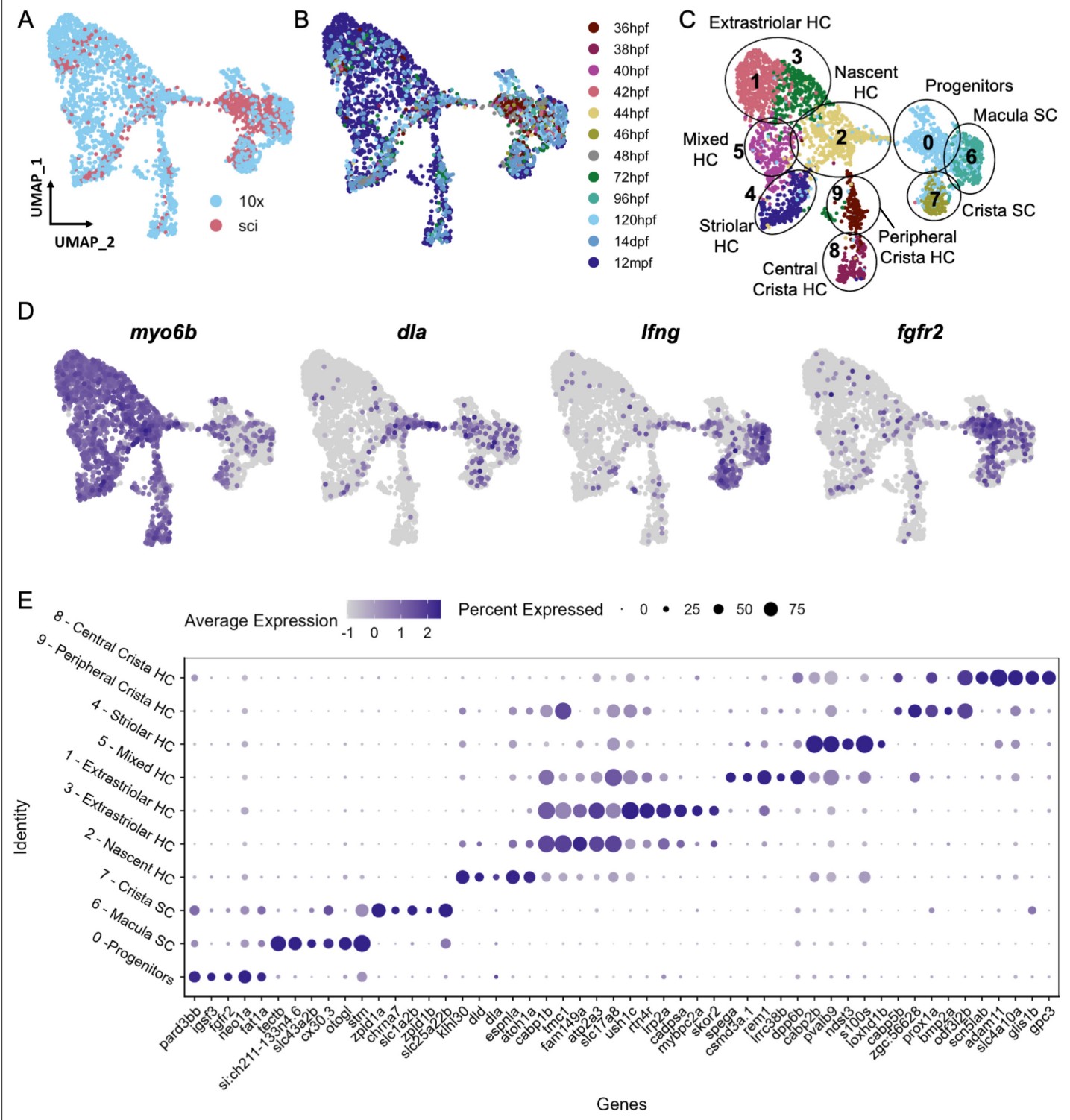

**Figure 3.** Cell subtypes in the zebrafish inner ear end organs. (**A–D**) Integration and analysis of single-cell RNAseq data generated by sci-Seq (sci) or 10x Chromium sequencing (10x) for inner ear hair cells and supporting cells from embryonic (sci), larval (sci,10x), and adult (10x) stages. UMAP projection of cells are grouped by (**A**) dataset of origin and (**B**) timepoint. (**C**) Unsupervised clustering divides cells into 10 clusters that were grouped into 9 cell subtypes. (**D**) Feature plots showing hair cell marker *myo6b*, nascent hair cell marker *dla*, supporting cell marker *lfng*, and putative progenitor marker *fgfr2* expression in the integrated dataset. (**E**) Differentially expressed genes across the 10 cell clusters.

The online version of this article includes the following figure supplement(s) for figure 3:

**Figure supplement 1.** scRNA-seq of 12 mpf zebrafish inner ear captures sensory hair cells and supporting cells as well as non-sensory supporting cells.

*Figure 3 continued on next page*

*Figure 3 continued*

**Figure supplement 2.** Hair cell and supporting cell marker expression in the integrated scRNA-seq dataset.

**Figure supplement 3.** Putative progenitor marker expression in individual progenitor and supporting cell clusters.

**Figure supplement 4.** Gene modules for integrated inner ear sensory patch dataset.

supporting cell trajectories (*fgfr2* and *igsf3*) and decreasing only along the hair cell trajectory (*fat1a* and *pard3bb*) (*Figure 4C and D*, *Figure 4—figure supplement 1B and C*). The hair cell trajectory progresses first through a stage marked by expression of *dla* and then *atoh1a* (Cluster 2, *Figure 4E*, *Figure 4—figure supplement 1D*). Concurrent with decreasing expression of nascent hair cell genes, we observe increasing expression of mature hair cell genes *gfi1aa* and *myo6b* (*Figure 4F*, *Figure 4— figure supplement 1E*). Along the supporting cell trajectory we observed upregulation of supporting cell-specific markers, including *stm* and *lfng* (*Figure 4G*, *Figure 4—figure supplement 1F*). These bifurcating lineage trajectories from Cluster 0 (*Figure 4A*) to hair and supporting cell clusters are consistent with the identification of Cluster 0 as a population of bipotent progenitors regulated by Notch signaling during early development (*Haddon et al., 1998*; *Riley et al., 1999*). To localize these developmental stages in vivo, we examined *dla* expression by in situ hybridization (*Figure 4—figure supplement 2*). We find that *dla* is expressed in supporting cells adjacent to myo6:GFP hair cells in both cristae and maculae, consistent with peripheral addition of new cells at the margins of the sensory patches.

## Distinct supporting cell types in the cristae versus maculae

Supporting cells comprise two major clusters that can be distinguished by expression of *tectb* and *zpld1a* among other genes (*Figure 3C*, see *Supplementary file 7* for differentially expressed genes). The *tectb* gene encodes Tectorin beta, a component of the tectorial membrane associated with cochlear hair cells in mammals (*Goodyear et al., 2017*), and a component of otoliths in zebrafish (*Kalka et al., 2019*). The z*pld1a* gene, encoding Zona-pellucida-like domain containing protein 1 a, is expressed in the cristae in fish (*Dernedde et al., 2014*; *Yang et al., 2011*) and mouse (*Vijaya-kumar et al., 2019*). Using fluorescent in situ hybridization, we find that *tectb* is expressed in the macular organs but not cristae, and *zpld1a* is expressed in cristae but not maculae (*Figure 5C and D*). Neither were detected in lateral line neuromasts (*Figure 5C and D*), showing they are inner ear-specific genes. Both *tectb* and *zpld1a* are expressed primarily in supporting cells, as they show little overlap in expression with the hair cell marker *myo6b*:GFP, similar to expression of the supporting cell marker *lfng* (*Figure 5B–D*, *Figure 5—figure supplement 1*). These results demonstrate the presence of distinct supporting cell subtypes for the maculae and cristae.

## Distinct types of hair cells in the zebrafish inner ear

While inner ear and lateral line hair cells share many structural and functional features, we sought to determine if these cells also have distinct molecular signatures. We compared published datasets of lateral line hair cells (*Baek et al., 2022*; *Kozak et al., 2020*; *Ohta et al., 2020*) to our data, restricting analysis to datasets generated by 10x Chromium preparation to avoid technical batch effects across studies. Using Scanorama for alignments (*Hie et al., 2019*), hair cells from the inner ear and lateral line form distinct clusters, with a number of differentially expressed genes (*Figure 2—figure supplement 3*), including the known markers for lateral line (*s100t*) and inner ear (*strc*) (*Figure 2*). This analysis suggests that inner ear hair cells of the maculae and cristae are more similar to each other than to lateral line hair cells.

Within the maculae and cristae, we find that hair cells can be subdivided into two major groups (clusters 1 and 3 versus cluster 4). These clusters are distinguished by differential expression of a number of genes including two calcium binding protein genes, *cabp1b* and *cabp2b* (*Di Donato et al., 2013*; *Figure 3E*). Hair cell cluster 5 has a mixed identity with co-expression of a number of genes shared between these two groups, including *cabp1b* and *cabp2b*.

We next tested the in vivo expression of genes in each cluster using in situ hybridization, choosing *cabp1b* and *cabp2b* as representative markers for each cluster (*Figure 6A*). In the larval cristae, utricle, and saccule, *cabp1b* and *cabp2b* mark *myo6b*+hair cells in largely non-overlapping zones (*Figure 6B–D*). By adult stages, complementary domains of *cabp1b*+and *cabp2b*+hair cells become

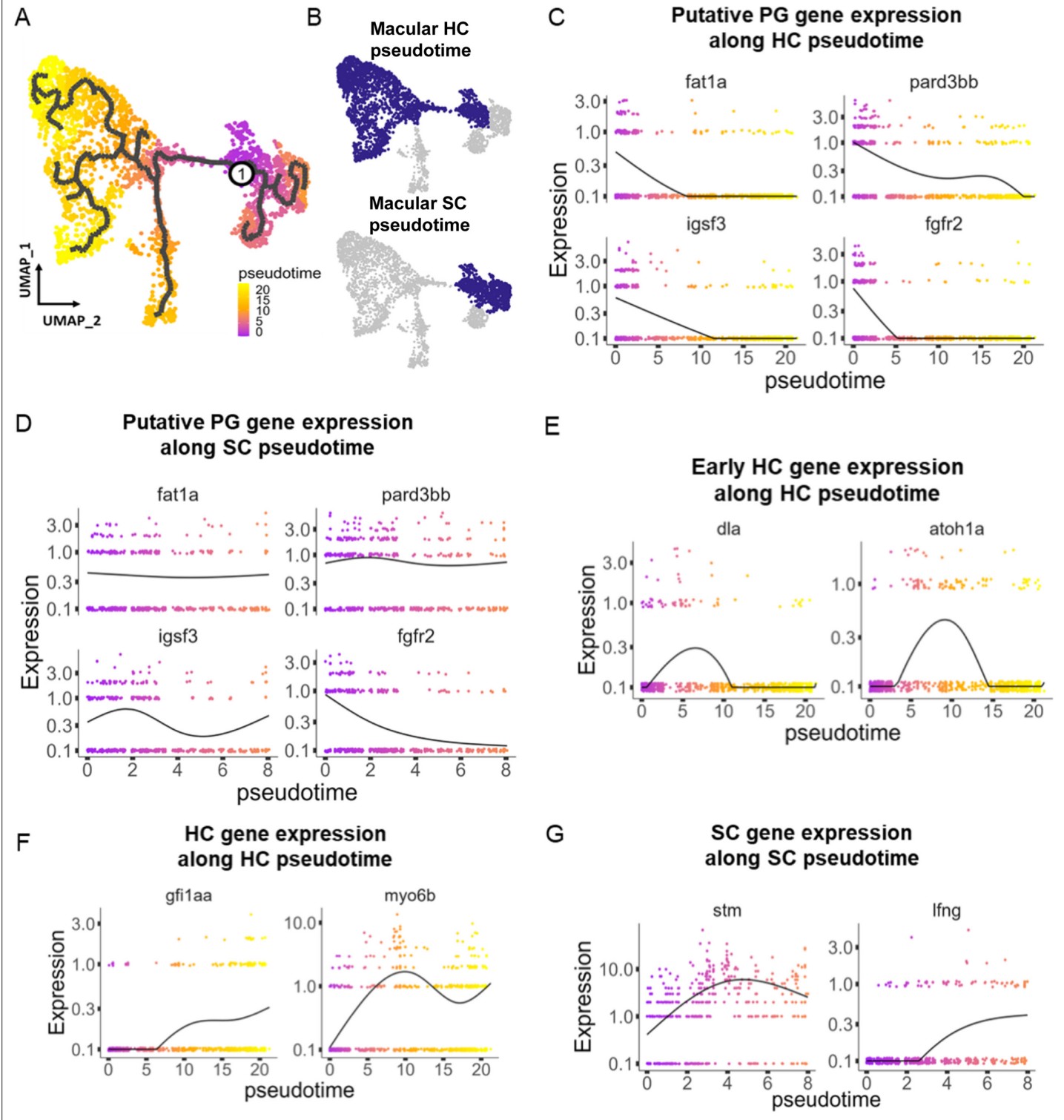

**Figure 4.** Pseudotime analysis reveals developmental trajectories in the zebrafish inner ear. (**A,B**) Pseudotime analysis of macular cells showing simulated developmental trajectories of a putative bipotent progenitor population into hair cell and supporting cell clusters. (**C,D**) Changes in putative progenitor markers along (**C**) hair cell and (**D**) supporting cell trajectories. *fat1a* and *pard3bb* only decrease along the hair cell trajectory, while *fgfr2* and *igsf3* decrease along both hair cell and supporting cell trajectories. (**E**) Transient expression of early hair cell genes *dla* and *atoh1a* along hair cell trajectories. (**F**) Increases in gene expression levels of *gfi1aa* and *myo6b* along hair cell trajectories. (**G**) Increases in *stm* and *lfng* along supporting cell trajectories.

*Figure 4 continued on next page*

clearly apparent (*Figure 6E–K*). In the adult utricle, a central crescent of *cabp2b+; myo6b*+hair cells is surrounded by a broad domain of *cabp1b*+; *myo6b*+hair cells. In the saccule and lagena, a late developing sensory organ, central *cabp2b+; myo6b*+hair cells are surrounded by peripheral *cabp1b+; myo6b*+hair cells. We also find several genes that are specific for hair cells in the cristae, utricle, or saccule (*Figure 7A*). These include the calcium binding protein gene *cabp5b* in the cristae, the transcription factor *skor2* in the utricle, and the deafness gene *loxhd1b* in the saccule (*Figure 7B–D*, *Figure 7—figure supplement 1*).

The domain organization of hair cells in the adult macular organs resembles that of striolar and extrastriolar hair cells in the mammalian utricle. We therefore examined expression of *pvalb9*, the zebrafish ortholog of the mouse striolar hair cell marker *Ocm* (*Hoffman et al., 2018*; *Jiang et al., 2017*; *Figure 8*, *Figure 8—figure supplement 1*). In the larval utricle, we observe near complete overlap of *pvalb9* with *cabp2b* (*Figure 8B–D*). In the adult utricle, there is substantial overlap of *pvalb9* with *cabp2b* expression (except for a thin strip of *pvalb9+; cabp2b-* cells), and little overlap with *cabp1b* expression (*Figure 8F and G*). In addition, anti-Spectrin staining of hair bundles reveals a line of polarity reversal within the *cabp2b*+domain of the utricle (*Figure 8H, I*), consistent with

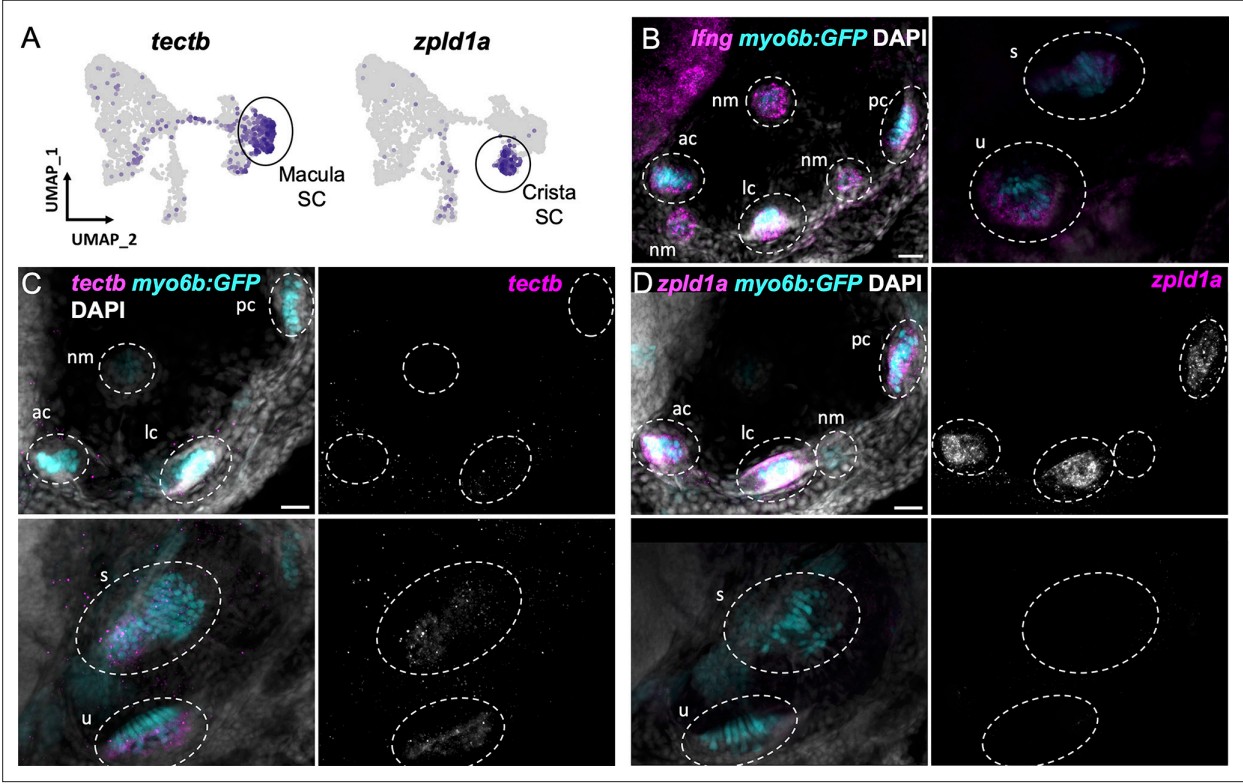

**Figure 5.** Distinct markers separate macula and crista supporting cells. (**A**) Feature plots showing expression of macula supporting cell marker *tectb* and crista supporting cell marker *zpld1a*. (**B–D**) HCR in situ hybridization in *myo6b*:GFP transgenic animals. Each set of images shown represents a projection of one z-stack split into cristae (lateral) and macula (medial) slices. Lateral line neuromasts positioned over the ear are visible in lateral slices. Expression pattern for (**B**) the pan-supporting cell marker *lfng*, (**C**) macula-specific marker *tectb*, and (**D**) crista-specific marker *zpld1a* in 5 dpf *myo6b*:GFP fish. Each set of images shown represents a projection of one z-stack split into cristae (lateral) and macula (medial) slices. ac: anterior crista, lc: lateral crista, nm: neuromast, pc: posterior crista, u: utricle, s: saccule. Scale bars = 20 μm.

The online version of this article includes the following figure supplement(s) for figure 5:

**Figure supplement 1.** *zpld1a* and *tectb* are primarily expressed in supporting cells.

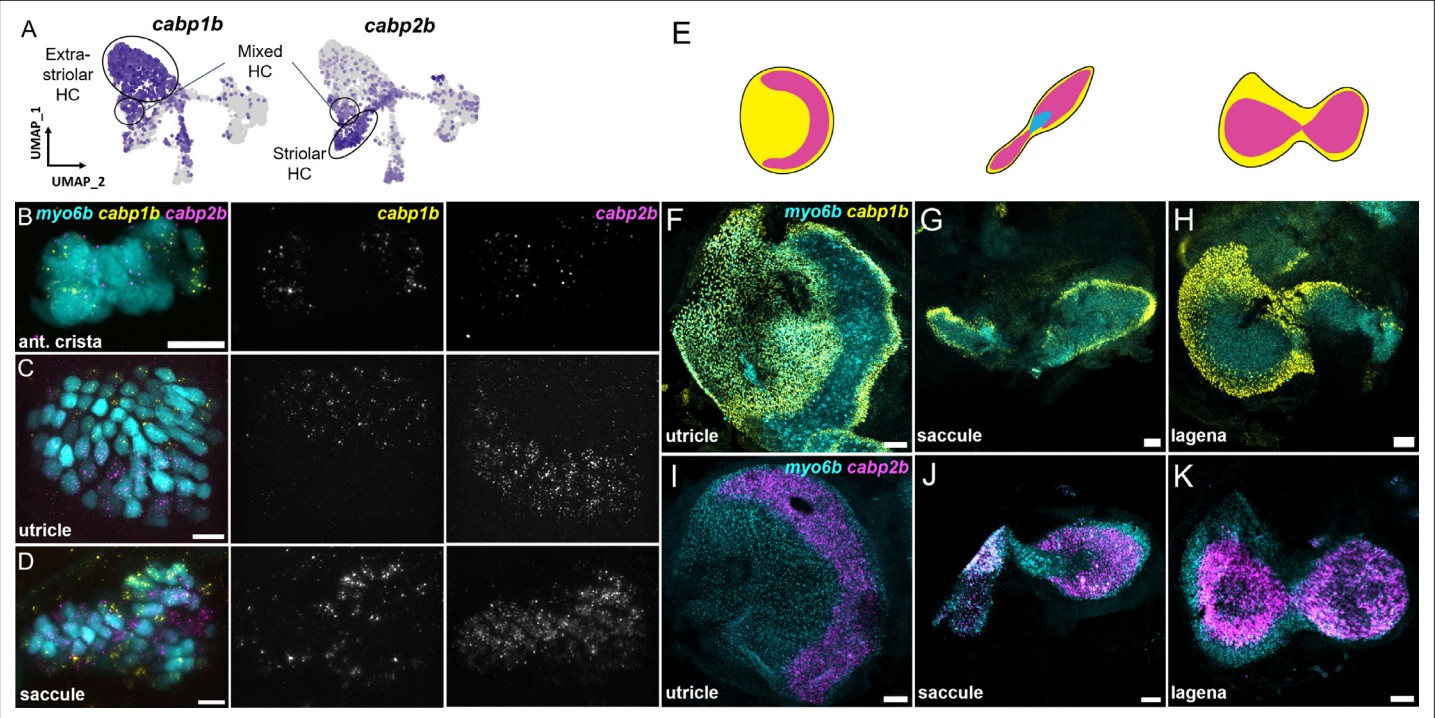

**Figure 6.** *cabp1b*+and *cabp2b*+label hair cells in distinct regions of sensory end organs. (**A**) Feature plots showing differential expression of *cabp1b* and *cabp2b* among crista and macula hair cells. (**B–D**) HCR in situ projections of individual sensory patches from 5 dpf *myo6b*:GFP fish showing differential spatial expression patterns of *cabp1b* and *cabp2b*. (**B**) *cabp1b* is expressed at the ends of the cristae, while *cabp2b* is expressed centrally. Anterior crista is shown. (**C**) In the utricle, *cabp1b* is expressed medially and *cabp2b* is expressed laterally. (**D**) In the saccule, *cabp1b* is expressed in peripheral cells at the dorsal and ventral edges of the organ. *cabp2b* is expressed centrally. Scale bars for HCR images = 10 µm. (**E**) Cartoon illustrations of the zebrafish utricle, saccule, and lagena, and the expression patterns of *cabp1b* (yellow) and *cabp2b* (magenta) within each sensory patch. (**F–H**) Wholemount RNAScope confocal images of adult inner ear organs showing peripheral expression pattern of *cabp1b* (n=3) in the adult zebrafish (**F**) utricle, (**G**) saccule, and (**H**) lagena. (**I–K**) Whole-mount RNAScope confocal images showing central expression pattern of *cabp2b* (n=4) in the adult zebrafish (**I**) utricle, (**J**) saccule, and (**K**) lagena. Scale bars for RNAScope images = 25 µm.

polarity reversal occurring within the striolar domains of mammalian macular organs (*Li et al., 2008*). Cluster 1/3 (*cabp1b*+) and Cluster 4 (*cabp2b*+) populations also differentially express genes related to stereocilia tip link and mechanotransduction channel components (*Figure 8—figure supplement 2*, *Supplementary file 8*) and various calcium and potassium channels (*Figure 8—figure supplement 3*, *Supplementary file 8*). We also note that the utricle marker *skor2* labels primarily extrastriolar hair cells within this end organ, with *loxhd1b* labeling striolar hair cells within the saccule. These findings suggest that zebrafish Cluster 4 (*cabp2b*+) and Cluster 1/3 (*cabp1b*+) hair cells largely correspond to striolar and extrastriolar hair cells, respectively, with distinct mechanotransduction and synaptic properties.

## Global homology of striolar and extrastriolar hair cells between fish and mice

To further probe similarities between zebrafish Cluster 4 (*cabp2b*+) and Cluster 1/3 (*cabp1b*+) hair cells versus striolar and extrastriolar hair cells in mammals, we utilized the Self-Assembling Manifold mapping (SAMap) algorithm (*Tarashansky et al., 2021*; *Musser et al., 2021*) to compare cell types across distant species. A strength of this algorithm is that it compares not only homologous gene pairs but also close paralogs, which is especially useful considering the extensive paralog switching observed between vertebrate clades (*Postlethwait, 2007*), as well as the extra round of genome duplication in the teleost lineage leading to zebrafish. When comparing adult zebrafish maculae with the postnatal mouse utricle (*Jan et al., 2021*), we find the highest alignment score between supporting cells (*Figure 9A*). Consistent with the spatial domains revealed by our in situ gene expression analysis, we find that mouse striolar Type I hair cells exclusively map to zebrafish Cluster 4 (*cabp2b*+)

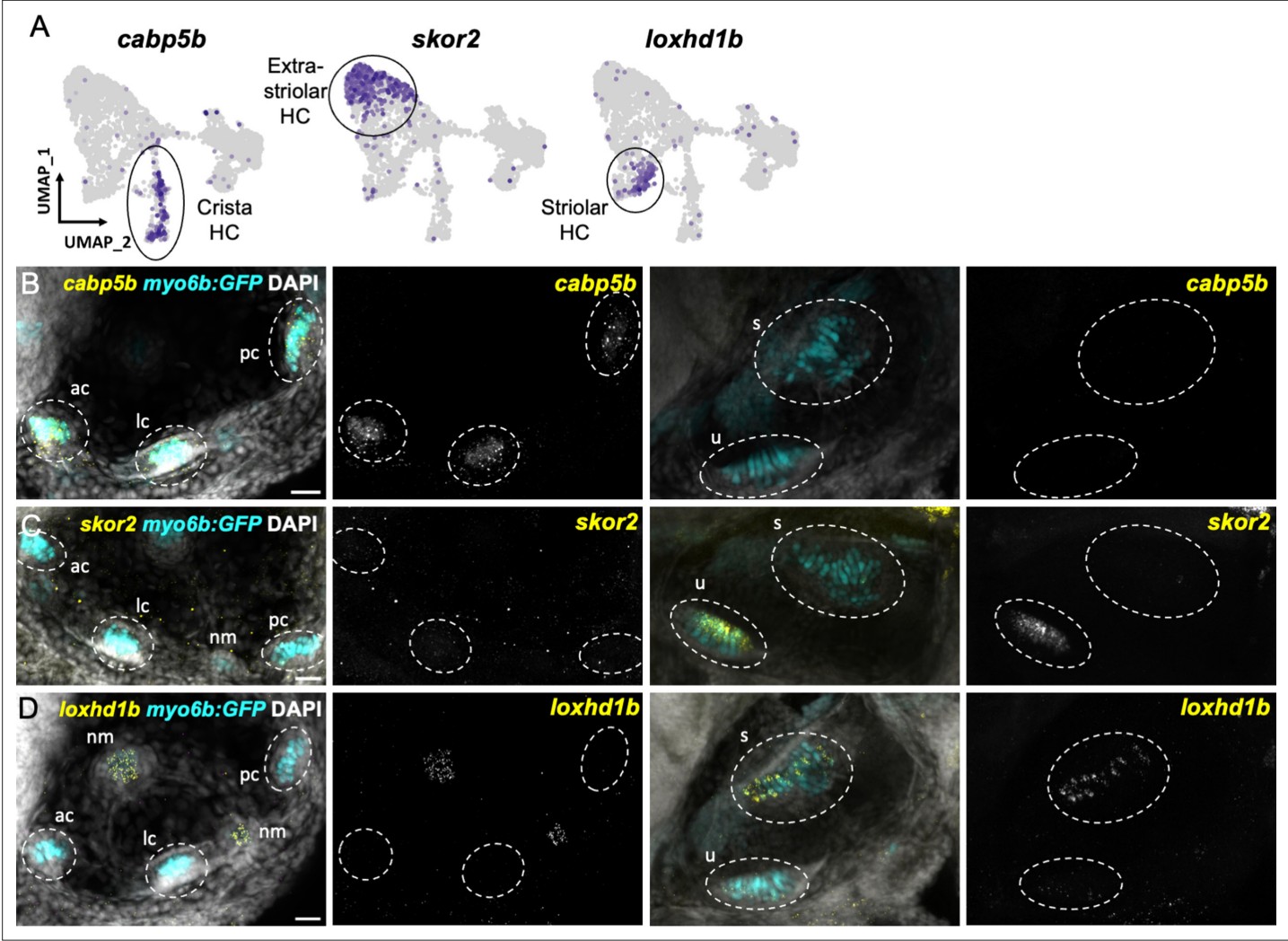

**Figure 7.** Distinct markers separate macula and crista hair cells. (**A**) Feature plots showing marker genes enriched in organ-specific subsets of inner ear hair cells: *cabp5b*, *skor2*, and *loxhd1b*. (**B–D**) HCR in situs in 5 dpf *myo6b*:GFP fish show expression of (**B**) *cabp5b* in crista but not macula hair cells, (**C**) *skor2* in the utricle only, and (**D**) *loxhd1b* in the saccule, as well as lateral line neuromast hair cells. Each set of images represents an orthogonal projection of one z-stack split into cristae (lateral) and macular (medial) slices. ac: anterior crista, lc: lateral crista, nm: neuromast, pc: posterior crista, s: saccule, u: utricle. Scale bar = 20 μm.

The online version of this article includes the following figure supplement(s) for figure 7:

**Figure supplement 1.** *skor2* and *loxhd1b* label subsets of hair cells in utricle or saccule.

hair cells, and mouse extrastriolar Type I and Type II hair cells predominantly to zebrafish Cluster 1/3 (*cabp1b*+) hair cells. In contrast, zebrafish lateral line hair cells (*Lush et al., 2019*) align exclusively to mouse extrastriolar and not striolar hair cells (*Figure 9—figure supplement 1*). The small degree of mapping of mouse extrastriolar Type I hair cells to zebrafish Cluster 4 (*cabp2b*+) hair cells suggests that zebrafish Cluster 4 (*cabp2b*+) hair cells may have more of a Type I identity than Cluster 1/3 (*cabp1b*+) cells in general. Gene pairs driving the homology alignment include striolar markers *Ocm*, *Loxhd1*, and *Atp2b2* for zebrafish Cluster 4 (*cabp2b*+) hair cells, and mouse extrastriolar markers *Tmc1*, *Atoh1*, and *Jag2* for zebrafish Cluster 1/3 (*cabp1b*+) hair cells (*Supplementary file 9*). Thus, zebrafish Cluster 4 (*cabp2b*+) macular hair cells are closely related to striolar cells of the mouse utricle, with zebrafish lateral line and Cluster 1/3 (*cabp1b*+) macular hair cells more closely related to mouse extrastriolar hair cells.

A recent single-cell study revealed distinct central versus peripheral hair cell subpopulations in postnatal mouse cristae, reminiscent of the striolar and extrastriolar populations in the maculae (*Wilkerson et al., 2021*). As our zebrafish cristae hair cells also separate into distinct clusters, Cluster

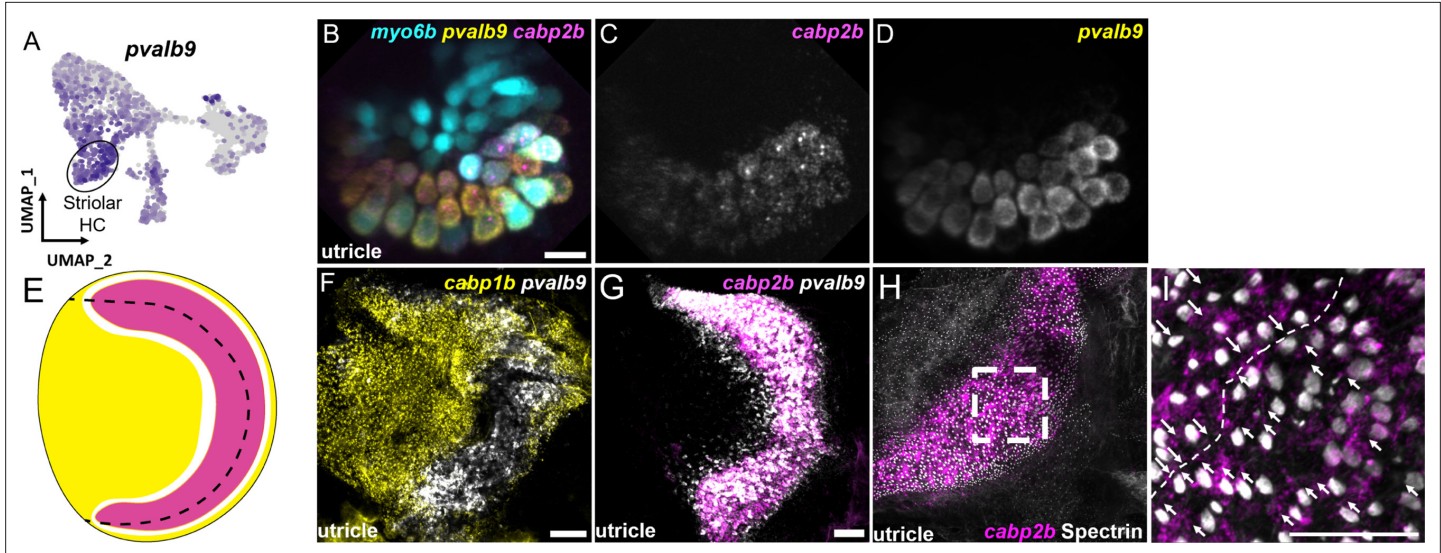

**Figure 8.** Zebrafish *cabp2b*+domain shares features with the mouse striolar region. (**A**) Feature plot shows enrichment for the striola marker *pvalb9* in *cabp2b*-expressing striolar cells. (**B–D**) HCR in situ in 5 dpf *myo6b*:GFP fish shows *pvalb9* and *cabp2b* co-expression in the utricle. Scale bar = 10 μm. (**E**) Cartoon illustration of overlapping expression of *pvalb9* (white) and *cabp2b* (magenta) that coincides with the line of hair cell polarity reversal. (**F, G**) Whole-mount RNAScope confocal images of adult zebrafish utricles showing expression of *pvalb9* relative to (**F**) *cabp1b* (n=3) and (**G**) *cabp2b* (n=4). Scale bar = 25 μm. (**H,I**) Whole-mount RNAScope RNA and protein co-detection assay showing co-localization of *cabp2b* expression (RNA) and the hair cell line of polarity reversal indicated by Spectrin (protein) staining (n=3). Scale bar = 25 μm. Arrows denote hair cell polarity and dotted line outlines line of polarity reversal.

The online version of this article includes the following figure supplement(s) for figure 8:

**Figure supplement 1.** Striola marker *pvalb9* is expressed in all inner ear sensory end organs.

**Figure supplement 2.** Inner ear hair cell subtypes differentially express mechanosensory apparatus genes.

**Figure supplement 3.** Inner ear hair cell subtypes differentially express voltage-gated calcium and potassium channel genes.

9 (*cabp1b+*) and Cluster 8 (*cabp2b+*) (*Figure 6A and B*), we performed SAMap analysis between the crista cell populations of the two species to investigate cell type homology. Similar to what we observed for the utricle, zebrafish centrally located Cluster 8 crista hair cells predominantly map to mouse central crista hair cells, and zebrafish peripherally located Cluster 9 crista hair cells exclusively map to mouse peripheral crista hair cells (*Figure 9B*, see *Supplementary file 10* for differentially expressed genes in Cluster 8 and Cluster 9 hair cells and *Supplementary file 11* for gene pairs driving homology). Conserved types of spatially segregated HCs therefore exist in both the maculae and cristae of zebrafish and mouse.

## Discussion

Our single-cell transcriptomic profiling of the embryonic to adult zebrafish inner ear reveals a diversity of hair cell and supporting cell subtypes that differ from those of the lateral line. As much of our knowledge about zebrafish hair cell regeneration comes from studies of the lateral line, understanding similarities and differences between the lateral line and inner ear has the potential to uncover mechanisms underlying the distinct regenerative capacity of inner ear hair cell subtypes. Recent tools to systematically damage inner ear hair cells in zebrafish (*Jimenez et al., 2021*) should enable such types of comparative studies.

We identify hair cells and supporting cells specific for maculae versus cristae, as well as two spatially segregated types of zebrafish inner ear hair cells with similarities to mammalian striolar and extrastriolar hair cells. These molecular signatures are conserved across larval and adult stages. However, consistent with other recent work (*Jimenez et al., 2022*; *Qian et al., 2022*), we were not able to resolve distinct clusters of hair cells or supporting cells corresponding to the distinct types of maculae: i.e. utricle, saccule, and lagena.

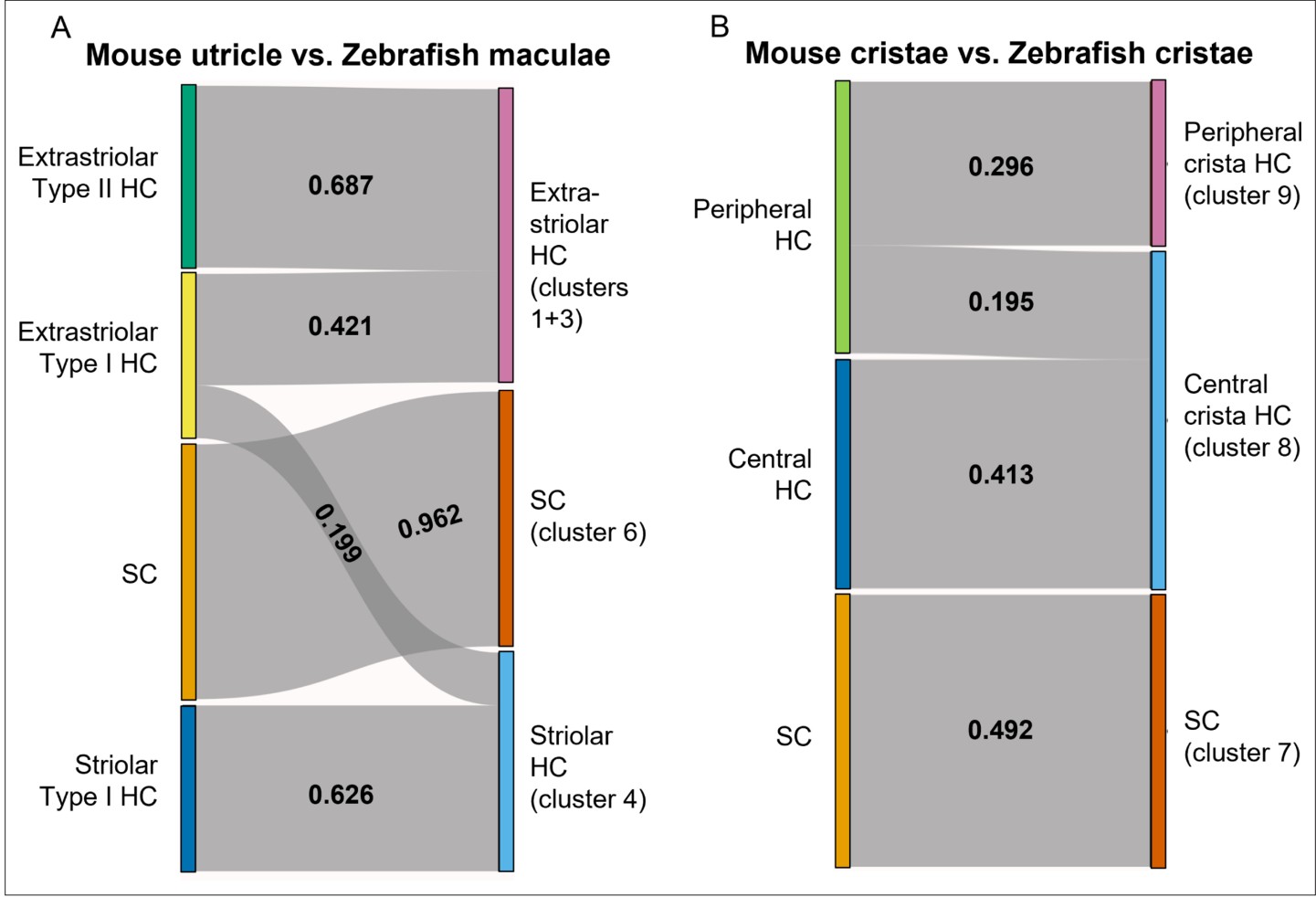

**Figure 9.** SAMap analysis reveals conserved gene expression patterns between mouse and zebrafish hair cell types. (**A–B**) Sankey plot showing the SAMap mapping scores (0–1) that indicate transcriptome relatedness between (**A**) mouse utricular and zebrafish macular single-cell clusters and (**B**) mouse and zebrafish cristae single-cell clusters. A mapping score of 0 indicates no evolutionary correlation in transcriptome while a mapping score of 1 indicates perfect correlation. Correlations below 0.15 were not plotted.

The online version of this article includes the following source data and figure supplement(s) for figure 9:

**Source data 1.** Mapping Scores between mouse utricle and zebrafish maculae hair and supporting cells.

**Source data 2.** Mapping Scores between mouse cristae and zebrafish cristae hair and supporting cells.

**Figure supplement 1.** SAMap analysis of mouse utricle versus zebrafish macular and lateral line cells.

**Figure supplement 1—source data 1.** Mapping Scores between mouse utricle and zebrafish maculae and lateral line hair cells.

The division of auditory and vestibular function across the otolith organs in zebrafish remains somewhat unclear. The saccule is thought to act as the primary auditory organ of larval zebrafish, as the utricle is not necessary for sound detection above low frequencies (*Yao et al., 2016*). In the zebrafish adult, excess sound exposure can damage the saccule, while damage to the utricle is unknown (*Schuck and Smith, 2009*). Conversely, the utricle is critical for larval vestibular function, while input from the saccule is unnecessary (*Riley and Moorman, 2000*). However, there is contrasting evidence for overlap in function of both saccule and utricle for sound detection in larvae (*Favre-Bulle et al., 2020*; *Poulsen et al., 2021*). Currently we are not able to identify clearly distinct hair cell types in the utricle compared to the saccule that might reflect functional differences; whether such genetic signatures exist remains an important question that will require further in-depth analysis. It is interesting to note that mammalian vestibular end organs are also capable of responding to high-frequency sound stimuli (reviewed in *Curthoys, 2017*), suggesting that sound detection by hair cells may not be linked to a distinct end organ-specific molecular signature.

Our study supports zebrafish possessing distinct types of striolar and extrastriolar hair cells in the maculae and cristae, with molecular differences between these subtypes implying different physiological properties. In the zebrafish utricle, vibration is preferentially transduced by striolar cells while static tilt is received by extrastriolar cells (*Tanimoto et al., 2022*). Consistent with use of a s100s-hs:tdTomato transgene to mark striolar cells in this previous study, s100s is a highly specific marker for our striolar hair cell cluster (Figure 3E). We also find zebrafish striolar and extrastriolar hair cell subtypes express distinct combinations of ion channel genes and mechanotransduction components, consistent with previous reports of distinct current profiles in central versus peripheral hair cells in the zebrafish utricle, saccule, and lagena (*Haden et al., 2013*; *Olt et al., 2014*), as well as spatial differences in ciliary bundle morphology and synaptic innervation in the larval zebrafish utricle (*Liu et al., 2022*). The distinct spatial distribution, channel expression, and hair bundle morphologies in these hair cells resembles the known spatial, electrophysiological, and hair bundle compositional differences seen in the striolar versus extrastriolar hair cells in the amniote vestibular end organs (*Holt et al., 2007*; *Kharkovets et al., 2000*; *Lapeyre et al., 1992*; *Meredith and Rennie, 2016*; *Moravec and Peterson, 2004*; *Rüsch et al., 1998*; *Xue and Peterson, 2006*).

In each of the zebrafish end organs, striolar and extrastriolar hair cells can be defined by differential expression of calcium binding proteins, in particular *cabp1b* versus *cabp2b*. As these calcium binding proteins closely interact with synaptic calcium channels (*Cui et al., 2007*; *Picher et al., 2017*) with potential functionally different consequences (*Yang et al., 2018*), their differential expression may confer unique electrophysiological properties to each cell type. Mutations in human *CABP2* associated with the autosomal recessive locus DFNB93 result in hearing loss (*Schrauwen et al., 2012*; *Picher et al., 2017*), underlining its functional importance. Even though we chose *cabp1b* and *cabp2b* as characteristic markers for zebrafish extrastriolar and striolar regions, it is worth noting that *Cabp2*, but not *Cabp1*, is expressed in all mouse postnatal utricular hair cells with differentially higher expression in the striola (*Jan et al., 2021*). Of note, lateral line hair cells express higher levels of *cabp2b* than *cabp1b* (*Lush et al., 2019*), despite our analysis suggesting that they are more closely related to extrastriolar hair cells. These observations emphasize the importance of examining global patterns of gene expression rather than individual markers when assigning homology of cell types.

By contrast, we found no clear homology of zebrafish inner ear hair cells with mammalian Type I and Type II hair cells. The lack of molecular signatures corresponding to Type I hair cells is consistent with previous reports that one of their major features, calyx synapses, are absent from macular organs in fishes (*Lysakowski and Goldberg, 2004*, but see *Lanford and Popper, 1996* for evidence of calyx synapses in goldfish cristae). These findings suggest that the diversification of inner ear hair cells into Type I and Type II cells may have largely emerged after the evolutionary split of ray-finned fishes from the lineage leading to mammals.

We recognize that identifying cell type homology across tissues and species through molecular analysis has several potential caveats. Although we have collected transcriptomic data from the zebrafish inner ear from a wide range of developmental stages, we are limited by the fact that the publicly available datasets for zebrafish lateral line and mouse utricle and cristae are restricted to immature stages. Thus, cell maturity could be a confounder in our analyses. However, when we limited the comparison of lateral line hair cells and postnatal mouse vestibular hair cells to 3–5 dpf inner ear hair cells, we see similar alignments as when we used our 12 mpf data (*Figure 9—figure supplement 1*). In addition, we collected fewer supporting cells from adult zebrafish than expected, skewing cell type representation towards hair cells (*Figure 3C*). Thus, additional optimization may be needed to further interrogate the cell subtypes within zebrafish inner ear supporting cell populations.

Nonetheless, our integrated dataset reveals distinct molecular characteristics of hair cells and supporting cells in the zebrafish inner ear sensory organs, with conservation of these patterns from larval stages to adults. Although not discussed in detail here, our data include additional cell populations of the zebrafish inner ear that express extracellular matrix-associated genes important for otic capsule structure and ion channel-associated genes associated with fluid regulation. These data form a resource that can be further explored to inform molecular aspects of hair cell electrophysiology, mechanotransduction, sound versus motion detection, maintenance of inner ear structure and ionic balance, and inner ear-specific hair cell regeneration.

## Materials and methods

### Zebrafish lines

This study was performed in strict accordance with the recommendations in the Guide for the Care and Use of Laboratory Animals of the National Institutes of Health. The Institutional Animal Care and Use Committees of the University of Southern California (Protocol 20771) and University of Washington (Protocol 2997–01) approved all animal experiments. Experiments were performed on zebrafish (*Danio rerio*) of AB or mixed AB/Tubingen background. For adult stages, mixed sexes of animals were used for constructing single-cell libraries, as well as RNAScope experiments. Published lines include *Tg(Mmu.Sox10-Mmu.Fos:Cre)*[zf384] (**Kague et al., 2012**); *Tg(–3.5ubb:LOXP-EGFP-STOP-LOXP-mCherry)*[cz1701Tg] (**Mosimann et al., 2011**); and *Tg(myosin 6b:GFP)*[w186] (**Hailey et al., 2017**).

### In situ hybridization and RNAScope

Hybridization chain reaction in situ hybridizations (Molecular Instruments, HCR v3.0) were performed on 5 dpf *myo6b*:GFP larvae as directed for whole-mount zebrafish embryos and larvae (**Choi et al., 2016**; **Choi et al., 2018**). Briefly, embryos were treated with 1-phenyl 2-thiourea (PTU) beginning at 24 hpf. At 5 dpf, larvae were fixed in 4% PFA overnight at 4 °C. Larvae were washed with PBS and then stored in MeOH at –20 °C until use. Larvae were rehydrated using a gradation of MeOH and PBST washes, treated with proteinase K for 25 min and post-fixed with 4% PFA for 20 min at room temperature. For the detection phase, larvae were pre-hybridized with a probe hybridization buffer for 30 min at 37 °C, then incubated with probes overnight at 37 °C. Larvae were washed with 5 X SSCT to remove excess probes. For the amplification stage, larvae were pre-incubated with an amplification buffer for 30 min at room temperature and incubated with hairpins overnight in the dark at room temperature. Excess hair pins were removed by washing with 5 X SSCT. Larvae were treated with DAPI and stored at 4 °C until imaging. All HCR in situ patterns were confirmed in at least three independent animals. Transcript sequences submitted to Molecular Instruments for probe generation are listed in *Supplementary file 12*. The *cabp1b* probes were tested on 3 separate occasions and imaged in at least 6 animals; *cabp2b* probes were tested on 5 separate occasions and imaged in at least 20 different animals; *cabp5b* probes were tested on 3 separate occasions and imaged in at least 9 different animals; *lfng* probes were tested on 2 separate occasions and imaged in at least 5 different animals; *loxhd1b* probes were tested on 2 separate occasions and imaged in at least 7 animals; *pvalb9* probes were tested on 2 separate occasions and imaged in at least 6 different animals; *skor2* probes were tested on 2 separate occasions and imaged in at least 6 different animals; *tectb* probes were tested on 4 separate occasions and imaged in at least 10 different animals; *zpld1a* probes were tested on 3 separate occasions and imaged in at least 9 different animals.

RNAScope samples were prepared by fixation in 4% paraformaldehyde either at room temperature for 2 hr or at 4 °C overnight. Adult (28–33 mm) inner ears were dissected and dehydrated in methanol for storage. RNAScope probes were synthesized by Advanced Cell Diagnostics (ACD): Channel 1 probe *myo6b* (1045111-C1), Channel 2 probe *pvalb9* (1174621-C2), and Channel 3 probes *cabp1b* (1137731-C3) and *cabp2b* (1137741-C3). Whole inner ear tissues were processed through the RNAScope Fluorescent Multiplex V2 Assay (ACD Cat. No. 323100) according to manufacturer's protocols with the ACD HybEZ Hybridization oven. *cabp1b* probe was tested on 4 separate occasions with 6 animals or 12 ears total; *cabp2b* probe was tested on 4 separate occasions with 7 animals or 14 ears total; *pvalb9* probe was tested on 2 separate occasions with 6 animals or 12 ears total. *myo6b* probe was used with each of the above probes.

### Immunofluorescence staining

Immediately following the RNAScope protocol, samples were prepared for immunofluorescence staining using mouse anti-β-Spectrin II antibody (BD Bioscience Cat. No. 612562, RRID: AB_399853). Briefly, RNAScope probed zebrafish ears were rehydrated in PBS for 5 min and rinsed in PBDTx (0.5 g bovine serum albumin, 500 μL DMSO, 250 μL 20% Triton-X in 50 mL PBS, pH = 7.4) for 15 min at room temperature. They were then blocked in 2% normal goat serum (NGS) in PBDTx for 3 hr at room temperature, and incubated with 1:500 dilution of mouse anti-β-Spectrin II antibody in PBDTx containing 2% NGS overnight at 4 °C. After three washes in PBDTx for 20 min each at room temperature, samples were incubated with 1:1000 dilution of Alexa 647 goat-anti-mouse IgG1 secondary antibody (Invitrogen Cat. No. A-21240, RRID: AB_2535809) for 5 hr at room temperature. They were then

washed 2 times in PBSTx (250 μL 20% Triton-X in 50 mL PBS) for 5 min each before imaging. Three animals or 6 ears total were subjected to Spectrin detection on 2 separate occasions.

## Imaging

Confocal images of whole-mount RNAScope samples were captured on a Zeiss LSM800 microscope (Zeiss, Oberkochen, Germany) using ZEN software. HCR-FISH imaging was performed on a Zeiss LSM880 microscope (Zeiss, Oberkochen, Germany) with Airyscan capability. Whole larvae were mounted between coverslips sealed with high vacuum silicone grease (Dow Corning) to prevent evaporation. Z-stacks were taken through the ear at intervals of 1.23 μm using a 10 X objective or through individual inner ear organs at an interval of 0.32 μm using a 20 X objective. 3D Airyscan processing was performed at standard strength settings using Zen Blue software.

## Single-cell preparation and analysis

### scRNA-seq library preparation and alignment

For 14 dpf animals (n=35), heads from converted *Sox10:Cre; ubb:LOXP-EGFP-STOP-LOXP-mCherry* fish were decapitated at the level of the pectoral fin with eyes and brains removed. For 12 mpf animals (n=6, 27–31 mm), utricle, saccule, and lagena were extracted from converted *Sox10:Cre; ubb:LOXP-EGFP-STOP-LOXP-mCherry* fish after brains and otolith crystals were removed. Dissected heads and otic sensory patches were then incubated in fresh Ringer's solution for 5–10 min, followed by mechanical and enzymatic dissociation by pipetting every 5 min in protease solution (0.25% trypsin (Life Technologies, 15090–046), 1 mM EDTA, and 400 mg/mL Collagenase D (Sigma, 11088882001) in PBS) and incubated at 28.5 °C for 20–30 min or until full dissociation. Reaction was stopped by adding 6×stop solution (6 mM CaCl2 and 30% fetal bovine serum (FBS) in PBS). Cells were pelleted (376 × g, 5 min, 4 °C) and resuspended in suspension media (1% FBS, 0.8 mM CaCl2, 50 U/mL penicillin, and 0.05 mg/mL streptomycin (Sigma-Aldrich, St. Louis, MO) in phenol red-free Leibovitz's L15 medium (Life Technologies)) twice. Final volumes of 500 μL resuspended cells were placed on ice and fluorescence-activated cell sorted (FACS) to isolate live cells that excluded the nuclear stain DAPI. For scRNAseq library construction, barcoded single-cell cDNA libraries were synthesized using 10 X Genomics Chromium Single Cell 3′ Library and Gel Bead Kit v.3.1 (14 dpf) or Single Cell Multiome ATAC +Gene Expression kit (12 mpf, single library built with all three sensory patches combined prior to library preparation, ATAC data not shown) per the manufacturer's instructions. Libraries were sequenced on Illumina NextSeq or HiSeq machines at a depth of at least 1,000,000 reads per cell for each library. Read2 was extended from 98 cycles, per the manufacturer's instructions, to 126 cycles for higher coverage. Cellranger v6.0.0 (10X Genomics) was used for alignment against GRCz11 (built with GRCz11.fa and GRCz11.104.gtf) and gene-by-cell count matrices were generated with default parameters.

### Data processing of scRNA-seq

Count matrices of inner ear and lateral line cells from embryonic and larval timepoints (18–96 hpf) were analyzed using the R package Monocle3 (v1.0.0) (*Cao et al., 2019*). Matrices were processed using the standard Monocle3 workflow (preprocess_cds, detect_genes, estimate_size_factors, reduce_dimension(umap.min_dist = 0.2, umap.n_neighbors = 25 L)). This cell data set was converted to a Seurat object for integration with 10 X Chromium sequencing data using SeuratWrappers. The count matrices of scRNA-seq data (14 dpf and 12 mpf) were analyzed by R package Seurat (v4.1.0) (*Hao et al., 2021*). Cells of neural crest origins were removed bioinformatically based on our previous study (*Fabian et al., 2022*). The matrices were normalized (NormalizeData) and integrated with normalized scRNA-seq data from the embryonic and larval time points according to package instruction (FindVariableFeatures, SelectIntegrationFeatures, FindIntegrationAnchors, IntegrateData; features = 3000). The integrated matrices were then scaled (ScaleData) and dimensionally reduced to 30 principal components. The data were then subjected to neighbor finding (FindNeighbors, k = 20) and clustering (FindClusters, resolution = 0.5), and then visualized through UMAP with 30 principal components as input. After data integration and processing, RNA raw counts from all matrices were normalized and scaled according to package instructions to determine gene expression for all sequenced genes, as the integrated dataset only contained selected features for data integration.

Mouse utricle scRNA-seq data (*Jan et al., 2021*) was downloaded from NCBI Gene Expression Omnibus (GSE155966). The count matrix was analyzed by R package Seurat (v4.1.0). Matrices were normalized (NormalizeData) and scaled for the top 2000 variable genes (FindVariableFeatures and ScaleData). The scaled matrices were dimensionally reduced to 15 principal components. The data were then subjected to neighbor finding (FindNeighbors, k = 20) and clustering (FindClusters, resolution = 1) and visualized through UMAP with 15 principal components as input. Hair cells and supporting cells were bioinformatically selected based on expression of hair cells and supporting cell markers *Myo6* and *Lfng*, respectively. Hair cells were further subcategorized into striola type I hair cells by co-expression of striola marker *Ocm* and type I marker *Spp*, extrastriola type I hair cells by expression of *Spp* without *Ocm*, and extrastriola type II hair cells by expression of *Anxa4* without *Ocm*.

Mouse crista scRNA-seq data (*Wilkerson et al., 2021*) was downloaded from NCBI Gene Expression Omnibus (GSE168901). The count matrix was analyzed by R package Seurat (v4.1.0). Matrices were normalized (NormalizeData) and scaled for the top 2000 variable genes (FindVariableFeatures and ScaleData). The scaled matrices were dimensionally reduced to 15 principal components. The data were then subjected to neighbor finding (FindNeighbors, k = 20) and clustering (FindClusters, resolution = 1) and visualized through UMAP with 15 principal components as input. Hair cells and supporting cells were bioinformatically selected based on expression of hair cell and supporting cell markers *Pou4f3* and *Sparcl1*, respectively. Hair cells were further subcategorized into central hair cells by expression of *Ocm* and peripheral hair cells by expression of *Anxa4*.

## Pseudotime analysis

We used the R package Monocle3 (v1.0.1) to predict the pseudo temporal relationships within the integrated scRNA-seq dataset of sensory patches from 36 hpf to 12 mpf. Cell paths were predicted by the learn_graph function of Monocle3. We set the origin of the cell paths based on the enriched distribution of 36–48 hpf cells. Hair (all macular hair cells, clusters 0–5) and supporting (macular supporting cells clusters 0 and 6) cell paths were selected separately (choose_cells) to plot hair cells and supporting cell marker expression along pseudotime (plot_genes_in_pseudotime).

## Differential gene expression

We utilized *presto* package's differential gene expression function to identify differentially expressed genes among the different cell types. Wilcox rank sum test was performed by the function *wilcox usc*. We then filtered for genes with log2 fold change greater than 0.5 and adjusted p-value less than 0.01. To compare inner ear hair cells to lateral line hair cells, we used the following datasets from GEO: 6–7 dpf lateral line hair cells (GSE144827, *Kozak et al., 2020*), 4 dpf lateral line hair cells (GSE152859, *Ohta et al., 2020*), and 5 dpf lateral line hair cells and supporting cells (GSE196211, *Baek et al., 2022*). Hair cells were selected from datasets by expression of *otofb* and integrated along with our 10 x Chromium dataset with Scanorama (*Hie et al., 2019*). Gene modules were computed in Monocle3 (v1.0.1) with a q-value cutoff of 1 x e-50.

## SAMap analysis for cell type homology

We used the python package SAMap (v1.0.2) (*Tarashansky et al., 2021*) to correlate gene expression patterns and determine cell type homology between mouse utricle (GSE155966) (*Jan et al., 2021*) or crista (GSE168901) (*Wilkerson et al., 2021*) hair cells and supporting cells and our 12 mpf zebrafish inner ear scRNA-seq data. Zebrafish lateral line hair cell sc-RNA data (GSE123241) (*Lush et al., 2019*) was integrated with our 12 mpf inner ear data using Seurat in order to compare to mice. First, a reciprocal BLAST result of the mouse and zebrafish proteomes was obtained by performing blastp (protein-protein BLAST, NCBI) in both directions using in-frame translated peptide sequences of zebrafish and mouse transcriptome, available from Ensembl (Danio_rerio.GRCz11.pep.all.fa and Mus_musculus.GRCm38.pep.all.fa). The generated maps were then used for the SAMap algorithm. Raw count matrices of zebrafish and mouse scRNA-seq Seurat objects with annotated cell types were converted to h5ad format using SeuratDisk package (v0.0.0.9020) and loaded into Python 3.8.3. Raw data were then processed and integrated by SAMap. Mapping scores between cell types of different species were then calculated by get_mapping_scores and visualized by sankey_plot. Gene pairs driving cell type homology were identified by GenePairFinder.

Single-cell RNA seq datasets are available from the NCBI Gene Expression Omnibus with Gene Set Accession number GSE211728.

## Acknowledgements

This manuscript is dedicated to Neil Segil, who was a wonderful colleague, friend, and mentor to many. We thank Megan Matsutani for fish care, the USC Stem Cell Flow Cytometry Core, and the CHLA Next-Generation Sequencing Core. We also thank David White and the UW Zebrafish Facility staff for fish care.

## Additional information

### Funding

| Funder | Grant reference number | Author |
|---|---|---|
| National Institute on Deafness and Other Communication Disorders | R21DC019948 | David W Raible |
| National Institute on Deafness and Other Communication Disorders | F31DC020898 | Marielle O Beaulieu |
| Hamilton and Mildred Kellogg Trust | | David W Raible |
| The Whitcraft Family Gift | | David W Raible |
| Hearing Health Foundation | | David W Raible |
| Paul G. Allen Frontiers Group | Allen Discovery Center for Cell Lineage Tracing | Cole Trapnell |
| National Human Genome Research Institute | UM1HG011586 | Cole Trapnell |
| National Human Genome Research Institute | 1R01HG010632 | Cole Trapnell |
| National Institute on Deafness and Other Communication Disorders | F31DC020633 | Tuo Shi |
| National Institute of Dental and Craniofacial Research | R35DE027550 | J Gage Crump |
| National Institute on Deafness and Other Communication Disorders | R01DC015829 | Neil Segil |
| National Institute on Deafness and Other Communication Disorders | T32DC009975 | Tuo Shi Neil Segil |
| National Institute on Deafness and Other Communication Disorders | T32DC005361 | Marielle O Beaulieu David W Raible |

The funders had no role in study design, data collection and interpretation, or the decision to submit the work for publication.

### Author contributions

Tuo Shi, Marielle O Beaulieu, Conceptualization, Data curation, Formal analysis, Validation, Investigation, Visualization, Methodology, Writing - original draft, Writing – review and editing; Lauren M Saunders, Resources, Software, Methodology; Peter Fabian, Resources, Methodology; Cole Trapnell, Resources, Software, Funding acquisition; Neil Segil, Conceptualization, Funding acquisition; J Gage Crump, Conceptualization, Supervision, Funding acquisition, Project administration, Writing – review

and editing; David W Raible, Conceptualization, Resources, Formal analysis, Supervision, Funding acquisition, Investigation, Project administration, Writing – review and editing

## Author ORCIDs

Tuo Shi (ID) http://orcid.org/0000-0002-5268-0146
Marielle O Beaulieu (ID) http://orcid.org/0000-0002-9819-2658
Lauren M Saunders (ID) http://orcid.org/0000-0003-4377-4252
Neil Segil (ID) http://orcid.org/0000-0002-0441-2067
J Gage Crump (ID) http://orcid.org/0000-0002-3209-0026
David W Raible (ID) http://orcid.org/0000-0002-5342-5841

## Ethics

This study was performed in strict accordance with the recommendations in the Guide for the Care and Use of Laboratory Animals of the National Institutes of Health. The Institutional Animal Care and Use Committees of the University of Southern California (Protocol 20771) and University of Washington (Protocol 2997-01) approved all animal experiments.

## Decision letter and Author response

Decision letter https://doi.org/10.7554/eLife.82978.sa1
Author response https://doi.org/10.7554/eLife.82978.sa2

# Additional files

## Supplementary files

- Supplementary file 1. Differentially expressed genes across inner ear and lateral line clusters.
- Supplementary file 2. PAGA scores for relative connectivity between clusters (related to *Figure 2—figure supplement 2*).
- Supplementary file 3. Gene modules for embryonic to larval inner ear and lateral line dataset.
- Supplementary file 4. Differentially expressed genes in inner ear cell clusters.
- Supplementary file 5. Gene modules for inner ear sensory patch dataset.
- Supplementary file 6. Genes enriched along pseudotime trajectories.
- Supplementary file 7. Genes enriched in supporting cell clusters.
- Supplementary file 8. Genes enriched in *cabp1b*+and *cabp2b*+macular cells.
- Supplementary file 9. Genes driving macular SAMap alignment.
- Supplementary file 10. Genes enriched in *cabp1b*+and *cabp2b*+crista cells.
- Supplementary file 11. Genes driving crista SAMap alignment.
- Supplementary file 12. cDNA sequences used for HCR in situ hybridization probes.
- MDAR checklist

## Data availability

Sequencing data have been deposited in GEO under accession code GSE211728.

The following dataset was generated:

| Author(s) | Year | Dataset title | Dataset URL | Database and Identifier |
|---|---|---|---|---|
| Shi T, Beaulieu MO, Saunders L, Fabian P, Trapnell C, Segil N, Crump JG, Raible DW | 2022 | Single-Cell Transcriptomic Profiling of the Zebrafish Inner Ear Reveals Molecularly Distinct Hair and Supporting Cell Subtypes | https://www.ncbi.nlm.nih.gov/geo/query/acc.cgi?acc=GSE211728 | NCBI Gene Expression Omnibus, GSE211728 |

The following previously published datasets were used:

| Author(s) | Year | Dataset title | Dataset URL | Database and Identifier |
|---|---|---|---|---|
| Kozak EL, Palit S, Miranda-Rodríguez JR, Janjic A, Böttcher A, Lickert H, Enard W, Theis F, López-Schier H | 2020 | Epithelial planar bipolarity emerges from Notch-mediated asymmetric inhibition of Emx2 | https://www.ncbi.nlm.nih.gov/geo/query/acc.cgi?acc=GSE144827 | NCBI Gene Expression Omnibus, GSE144827 |
| Ohta S, Martin D, Wu D, Ji YR | 2020 | Emx2 defines bidirectional polarity of neuromasts by changing hair-bundle orientation and not hair-cell positions | https://www.ncbi.nlm.nih.gov/geo/query/acc.cgi?acc=GSE152859 | NCBI Gene Expression Omnibus, GSE152859 |
| Baek S, Tran NT, Diaz DC, Tsai Y, Piotrowski T | 2022 | High-resolution single cell transcriptome analysis of zebrafish sensory hair cell regeneration | https://www.ncbi.nlm.nih.gov/geo/query/acc.cgi?acc=GSE196211 | NCBI Gene Expression Omnibus, GSE196211 |
| Jan TA, Eltawil Y, Ling AH, Chen L, Ellwanger D, Heller S, Cheng AG | 2021 | Single cell RNA seq analysis of postnatal mouse utricle | https://www.ncbi.nlm.nih.gov/geo/query/acc.cgi?acc=GSE155966 | NCBI Gene Expression Omnibus, GSE155966 |
| Wilkerson BA, Zebroski HL, Finkbeiner CR, Chitsazan AD, Beach KE, Sen N, Zhang RC, Bermingham-McDonogh O | 2021 | Single-cell Transcriptomic Analysis of the Mouse Crista Ampullaris | https://www.ncbi.nlm.nih.gov/geo/query/acc.cgi?acc=GSE168901 | NCBI Gene Expression Omnibus, GSE168901 |
| Lush ME, Diaz DC, Koenecke N, Baek S, Boldt H, St Peter MK, Gaitan-Escudero T, Romero-Carvajal A, Busch-Nentwich EM, Perera AG, Hall KE, Peak A, Haug JS, Piotrowski T | 2019 | Single cell RNA-Seq reveals Fgf signaling dynamics during sensory hair cell regeneration | https://www.ncbi.nlm.nih.gov/geo/query/acc.cgi?acc=GSE123241 | NCBI Gene Expression Omnibus, GSE123241 |
| Fabian P, Tseng KC, Thiruppathy M, Arata C, Chen HJ, Smeeton J, Nelson N, Crump JG | 2022 | Single-cell profiling of cranial neural crest diversification across a vertebrate lifetime | https://www.ncbi.nlm.nih.gov/geo/query/acc.cgi?acc=GSE178969 | NCBI Gene Expression Omnibus, GSE178969 |

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
