## [Editor Report]

This important study describes transcriptomic profiles of sensory and non-sensory cells of the zebrafish inner ear at single-cell resolution in embryonic through adult stages. These solid results catalogue transcriptomic data and show evidence that distinct cell subtypes exist between cells of the ear and the lateral line as well as within subcellular compartments in the inner ear. These findings provide information towards comparative studies of inner ear hair cell function and regeneration.

---

## [Decision Letter]

**Decision letter after peer review:**

Thank you for submitting your article "Single-Cell Transcriptomic Profiling of the Zebrafish Inner Ear Reveals Molecularly Distinct Hair Cell and Supporting Cell Subtypes" for consideration by *eLife*. Your article has been reviewed by 3 peer reviewers, including Lavinia Sheets as the Reviewing Editor and Reviewer #1, and the evaluation has been overseen by a Reviewing Editor and Didier Stainier as the Senior Editor. The following individual involved in the review of your submission has agreed to reveal their identity: Shawn M Burgess (Reviewer #3).

Essential revisions:

1) Address concerns about how the expression and function of Cabp1 & 2 in mammals corresponds with the observations made in zebrafish.

2) Address reviewer-specific comments on the content of the manuscript without further experiments.

*Reviewer #1 (Recommendations for the authors):*

– Introduction:

The statement "In larval zebrafish, both saccule and utricle hair cells respond to sound stimuli of frequencies between 100-4000 Hz" is not accurate and is not supported by the studies the authors cited. Several specific points: (1) none of the studies cited tested frequencies above 1000 Hz (2) microphonic potential recordings in larvae (Yao et al., 2016) showed that microphonic thresholds increased particularly at low frequencies when the utricular otolith was displaced (i.e. removed) and high frequencies when the saccular otolith was displaced, indicating differences in hair cell frequency tuning (3) numerous behavior studies in larval zebrafish have shown that the utricle is the gravity sensing organ of the fish and maximally sensitive to lower frequencies (4) studies in larval and adult zebrafish support that the saccule is more sensitive to higher frequencies, contributes to a sound-evoked startle response, and can be damaged by excess sound. A few suggested studies for your reference: Mo et al., BMC Neuroscience, 2010; Bagnall & Schoppik, Current Op in Neuroscience, 2018; Bhandiwad, JARO, 2018; Smith, Hearing Research, 2009

– Results and Discussion

That inner ear hair cells and supporting cells are distinct from those of the lateral line is an interesting observation but is only examined until 96 hpf, when the lateral line is still reaching functional maturation. Do the authors predict similar segregation of expression profiles in more mature lateral-line organs? This seems like an important distinction, given the subsequent SAMap analysis with data from 5 dpf larvae.

Given that the saccule and the utricle in zebrafish have different frequency selectivity and specific functional roles, it would be informative to understand how their expression profiles are different. Further, even though hearing is discussed at the beginning of this study, conservation between expression profiles observed in hair cells and supporting cells of the cochlea are not examined. This seems like an oversight given that these organs (particularly the saccule) are known to be used for hearing in fish.

Clarification and more in-depth discussion on the significance of differential capb1b and capb2b expression in the zebrafish inner ear are needed. The expression data in this study suggests that cabp2b + cells are related to striolar cells of the mouse utricle while cabp1b + are more closely related to extrastriolar cells. Yet these observations regarding Cabp1 & 2 do not correspond with what is reported from expression studies in the mammalian utricle (gEAR database). Further, the database of gene expression in 5 dpf lateral line hair cells reported by Lush et al. shows high expression of cabp2b in lateral line hair cells and minimal expression of cabp1b, contradicting the conclusion that lateral line hair cells are more closely related to extrastriolar/ cabp1b + cells. In mammals, Cabp2 is known to inhibit presynaptic CaV1.3 inactivation, which is critical for sound encoding, while Cabp1 is important for the proper function of auditory nerves (see Yang et al., Hearing Research, 2018). The different expression patterns reported here in zebrafish are striking; contextualizing these observations with what is known about the expression and function of these genes in mammals would provide more useful context for hearing and balance researchers.

SAMp analysis – it appears expression data from adult fish ears (12 mpf) was integrated with lateral line neuromast data from 5 dpf larvae. Given that the properties of hair cells are functionally more mature in older fish (Olt et al., J Physiol., 2016) is this a valid comparison?

*Reviewer #2 (Recommendations for the authors):*

Overall the data are nicely presented and there is some limited validation of new markers. The presence of striolar and extrastriolar regions of the utricle and saccule are demonstrated, although this is, perhaps, not that surprising given that striolar and extrastriolar regions of these structures have been reported in other fish species. My primary hesitation with this study is whether the overall level of advancement is sufficient for *eLife*? Single-cell data sets are now quite common, so I have to wonder what the advance is here.

Specific comments:

Page 5, 3rd paragraph: comment about otolith crystals should be clarified as fish have just single otoliths while mammals have multiple smaller crystals.

Page 7, I'm unclear on the usefulness of this first analysis, in particular regarding non-sensory inner ear cells. Known markers of the inner ear and lateral line non-sensory cells were used to identify individual cells within the data set. So wouldn't this bias the collection of cells towards those that are different? And if this is the case, is it remarkable that the cells don't overlap given that differential gene expression was used to select them?

This would not appear to be the case for the comparison of hair cells which were selected using pan-hair cell markers.

Page 10, end of the first paragraph: given all the regeneration data and analysis of notch mutants, it seems like a pretty safe bet that HCs and SCs arise from a common progenitor, even without lineage tracing data.

Figure 3: the data set seemed to be highly skewed towards hair cells with relatively few non-sensory cells collected. Is this the case, or are there more hair cells than non-sensory cells?

Figure 3D: it would be helpful to include a feature plot for *atoh1a* as dla does not intuitively appear to be a nascent hair cell marker given the large number of progenitors that also express this transcript.

Figure 4B: so are the crista HCs and crista SCs not included in the trajectory?

*Reviewer #3 (Recommendations for the authors):*

Review of the Results and Figures

The authors' main goal in the manuscript is to understand the diversity in hair cell and support cell subtypes in the zebrafish inner ear and their relationship to the mammalian inner ear. Overall, the manuscript's readability can be revised to improve the flow of the text, but there is sufficient information presented for readers to follow the rationale, procedures, and insights. The results are clear, and the methods are appropriate.

Some recommendations to strengthen the readability and conclusions of the manuscript:

The authors find that hair cells and support cells in the inner ear are transcriptionally distinct from the lateral line systems.

Figure 1 should include a schematic of the lateral line systems. It should also be described in the figure legend that the zebrafish and mouse inner ears represent adult structures.

There are 2 paragraphs in the Results section that describe inner ear vs lateral line results on page 7 (of the merged file) and page 11. The results related to the distinct molecular signatures of the lateral line and inner ear should be moved earlier in the text and combined with the results on page 7 to improve the flow of the text. Figure S5 is a great figure showing that the lateral line and inner ear clusters do not overlap which indicates their differences, but this figure is redundant since a similar UMAP is shown in Figure 2B. If the Results section regarding the differences between the lateral line and inner ear are combined, then the authors can also combine Figure S5B with Figure 2. Tables of marker gene data for each partition related to Figure 2B and Figures 2C-F should be provided as a supplement and indicated in the Results section of the text.

Figure 2. The colors along the horizontal axes in Figure 2C-2F correspond to the colors in Figure 2B, but they should be clearly described in the figure legend for readers.

The authors identify distinct hair cells and supporting cell types in juvenile and adult inner ears of zebrafish. They also identify an additional putative progenitor cluster.

Figure S2 represents UMAPs of 12 mpf zebrafish inner ear. However, this is the first and only time we are seeing these UMAPs of 12 mpf ears exclusively. It would be beneficial to the reader if there was a UMAP of the unsupervised clustering adjacent to the feature plots. Labels should be included to guide the reader to the structural cells that the author is describing. OR, the authors can remove this figure as it doesn't contribute new information and appears to be a processing step that can be described in the methods section.

Figure S3 represents feature plots of genes expressed in the integrated scRNA-seq dataset and clearly displays hair cell genes and support cell genes. The progenitor cell markers selected for this figure do not clearly show that the transcriptional signature of the putative progenitor cell type is distinct from the other cell types. From Figure S3, it appears that the pan-supporting markers overlap with putative progenitor markers. Are there other markers in the progenitor pool from Table S2 that can be selected to display their transcriptional distinctiveness? The authors should draw an outline or point to where the progenitor cell types are in the UMAP plots.

Figure 3 also displays feature plots of genes expressed in hair cells, support cells, and progenitor cell types. The authors state that the putative progenitor cells show weak expression of hair cells and support cell markers. However, according to Figure 3D, macula and cristae support cells have high expression of the progenitor cell gene *Fgfr2*. Visually, the observation they write about isn't clear. It might be helpful to include the fold change in gene expression relative to the other clusters to support their observation that the novel progenitor cell genes are enriched.

The authors propose that the progenitor cells represent a transition state cell type that may contribute to either hair cells or support cells which is also supported by recent work on regenerating zebrafish inner ears. Although lineage tracing is required to validate this idea, the authors explore cell fate trajectories using Monocle3 to examine gene expression as differentiation progresses. They provide the Morans I test results in Table S3, but it would be more informative if the authors clustered genes into modules that are co-expressed across cells. An additional layer of Monocle3 analysis (i.e. gene_module_df) will uncover novel regulatory interactions governing the support cell to progenitor cell transition and the progenitor cell to hair cell transition.

Considering that the authors have beautifully labelled hair cell and support cell subtypes to validate their transcriptomic findings, can they also label progenitor cells at the single cell level with spatial information in the inner ear tissues using in situ hybridization assays?

The authors carefully describe support cell types in cristae versus maculae. According to the methods section, the authors performed single-cell experiments on the lagena. What are the differences between the lagena, saccule, and utricle? Can information about the lagena be extracted from the integrated analysis?

For Figure 5, it would be helpful to have a schematic of the zebrafish 5dpf inner ear anatomy with labels. Alternatively, a schematic of the 5dpf inner ear can be added to Figure 1.

Similarly, Figure 6 can include schematics of the adult saccule and utricle OR these structures could be shown in Figure 2. In Figure 6, the utricle, saccule, and lagena should be outlined and have arrows to guide the reader. Figure 8 should also include outlines of the utricles.

Review of the Methods Section

For Single-cell preparation and analysis, it should be clear whether separate libraries for lagena, saccule, and utricle were prepared or if the dissected structures were combined before library preparation.

---

## [Author Response]

Reviewer #1 (Recommendations for the authors):– Introduction:The statement "In larval zebrafish, both saccule and utricle hair cells respond to sound stimuli of frequencies between 100-4000 Hz" is not accurate and is not supported by the studies the authors cited. Several specific points: (1) none of the studies cited tested frequencies above 1000 Hz (2) microphonic potential recordings in larvae (Yao et al., 2016) showed that microphonic thresholds increased particularly at low frequencies when the utricular otolith was displaced (i.e. removed) and high frequencies when the saccular otolith was displaced, indicating differences in hair cell frequency tuning (3) numerous behavior studies in larval zebrafish have shown that the utricle is the gravity sensing organ of the fish and maximally sensitive to lower frequencies (4) studies in larval and adult zebrafish support that the saccule is more sensitive to higher frequencies, contributes to a sound-evoked startle response, and can be damaged by excess sound. A few suggested studies for your reference: Mo et al., BMC Neuroscience, 2010; Bagnall & Schoppik, Current Op in Neuroscience, 2018; Bhandiwad, JARO, 2018; Smith, Hearing Research, 2009.

We apologize for inadvertently leaving out the reference to Poulsen et al. 2021 Current Biology, demonstrating zebrafish larvae neural responses to sound stimuli up to 4 kHz. We agree that the relative contribution of utricle and saccule to hearing is an interesting one, related to our observations that these end organs are not well-distinguished by our scRNA-seq analysis. Although there is some evidence supporting a clear distinction between saccule and utricle with respect to hearing, particularly across fishes, we note that there is also good evidence for both of these end organs involved in hearing in zebrafish. We now include a more thorough analysis of this issue in the introduction (lines 73-77) and discussion (lines 316-334).

– Results and DiscussionThat inner ear hair cells and supporting cells are distinct from those of the lateral line is an interesting observation but is only examined until 96 hpf, when the lateral line is still reaching functional maturation. Do the authors predict similar segregation of expression profiles in more mature lateral-line organs? This seems like an important distinction, given the subsequent SAMap analysis with data from 5 dpf larvae.

We agree that this is a potential caveat for our analysis, and now highlight this issue in the discussion (lines 371-381). However we note that the specific comparison of hair cells between lateral line and inner ear (Figure 2—figure supplement 3) included data from hair cells isolated from 6-7 dpf larvae (Kozak et al., 2020). We also note that the available mouse data for which we performed the SAMap analysis is from early postnatal tissue, so it is not readily apparent that it is just a matter of immature vs mature samples driving cross-species differences. Although our own inner ear data cover extensive stages from embryonic to adult stages, we are limited by lack of adult lateral line and adult mouse inner ear data.

Given that the saccule and the utricle in zebrafish have different frequency selectivity and specific functional roles, it would be informative to understand how their expression profiles are different. Further, even though hearing is discussed at the beginning of this study, conservation between expression profiles observed in hair cells and supporting cells of the cochlea are not examined. This seems like an oversight given that these organs (particularly the saccule) are known to be used for hearing in fish.

We attempted to distinguish the transcriptome between saccular and utricular hair cells with our datasets. However, we were only able to identify and verify a handful of genes that are differentially expressed between these two end organs (Figure 7). The lack of drastic molecular differences between the two agrees with findings from Jimenez et al. 2022, where they built single-cell libraries separately from the utricle and saccules. This suggests that the expression profiles of these two sensory patches are more similar to each other than one might predict.

We agree that it would be insightful to further compare zebrafish otic hair cells to mouse cochlear hair cells. Unfortunately, the publicly available mouse cochlear dataset contained a limited number of hair cells, which has been a challenge for us in preliminary analysis to identify clear correlations between gene expression profiles of zebrafish otic and mouse cochlear hair cells. Full analysis awaits the availability of more comprehensive mouse data.

Clarification and more in-depth discussion on the significance of differential capb1b and capb2b expression in the zebrafish inner ear are needed. The expression data in this study suggests that cabp2b + cells are related to striolar cells of the mouse utricle while cabp1b + are more closely related to extrastriolar cells. Yet these observations regarding Cabp1 & 2 do not correspond with what is reported from expression studies in the mammalian utricle (gEAR database). Further, the database of gene expression in 5 dpf lateral line hair cells reported by Lush et al. shows high expression of cabp2b in lateral line hair cells and minimal expression of cabp1b, contradicting the conclusion that lateral line hair cells are more closely related to extrastriolar/ cabp1b + cells. In mammals, Cabp2 is known to inhibit presynaptic CaV1.3 inactivation, which is critical for sound encoding, while Cabp1 is important for the proper function of auditory nerves (see Yang et al., Hearing Research, 2018). The different expression patterns reported here in zebrafish are striking; contextualizing these observations with what is known about the expression and function of these genes in mammals would provide more useful context for hearing and balance researchers.

We agree that these issues are worth more discussion and have added this (lines 355-362), including the noted differences between zebrafish inner ear, lateral line and mouse. We now clarify that we used cabp1b and cabp2b as characteristic markers for the expression clusters because of their relatively strong levels of expression and differential distribution in the inner ear rather than as specific probes for function, and we now clarify that we are comparing clusters throughout the Results section. We also note that comparisons between tissues incorporate the whole gene expression profile of a cluster, not just a single marker. So while cabp1b may be expressed in the lateral line, the other genes differentially expressed across clusters are not. We also include more discussion of functional roles for these calcium binding proteins (lines 350-355).

SAMp analysis – it appears expression data from adult fish ears (12 mpf) was integrated with lateral line neuromast data from 5 dpf larvae. Given that the properties of hair cells are functionally more mature in older fish (Olt et al., J Physiol., 2016) is this a valid comparison?

We agree that the potential stage differences between tissues is important to acknowledge and now discuss this caveat in the discussion (lines 371-381). However we note that the SAMap analysis does not directly compare lateral line to zebrafish inner ear but rather to mouse inner ear. Perhaps a more relevant discrepancy is that the available mouse data is early postnatal, when the ear is clearly immature. To address this issue we limited analysis to zebrafish inner ear from 3 and 5 dpf larvae and 5 dpf neuromast and performed the same SAMap analysis to mouse utricle. The resulting alignment is similar to what we have found with adult zebrafish ear data. We now include this new SAMap analysis as Figure 9—figure supplement 1.

Reviewer #2 (Recommendations for the authors):Overall the data are nicely presented and there is some limited validation of new markers. The presence of striolar and extrastriolar regions of the utricle and saccule are demonstrated, although this is, perhaps, not that surprising given that striolar and extrastriolar regions of these structures have been reported in other fish species. My primary hesitation with this study is whether the overall level of advancement is sufficient for eLife? Single-cell data sets are now quite common, so I have to wonder what the advance is here.Specific comments:

Page 5, 3rd paragraph: comment about otolith crystals should be clarified as fish have just single otoliths while mammals have multiple smaller crystals.

We have edited the text to clarify that fish end organs have single otoliths. We have also added a schematic of the larval zebrafish inner ear to Figure 1, which includes the utricular and saccular otoliths.

Page 7, I'm unclear on the usefulness of this first analysis, in particular regarding non-sensory inner ear cells. Known markers of the inner ear and lateral line non-sensory cells were used to identify individual cells within the data set. So wouldn't this bias the collection of cells towards those that are different? And if this is the case, is it remarkable that the cells don't overlap given that differential gene expression was used to select them?This would not appear to be the case for the comparison of hair cells which were selected using pan-hair cell markers.

We now clarify how we selected inner ear and lateral line cells from the large 1.25 million cell dataset of Saunders et al. Cell clusters were first identified by expression of *eya1*, which is broadly expressed in lateral line and inner ear cells. This set of 16 thousand cells was then re-analyzed by unsupervised clustering, and the identities of these new clusters confirmed by known marker genes. At no point did we filter out individual cells by specific gene expression, rather we used the entire group segregated by unsupervised clustering.

It is difficult to predict how similar two groups of cells will be based on the expression of a handful of marker genes. It is interesting that the ear nonsensory cells express many ear-specific genes at relatively high levels (e.g. stm, otomp), while the lateral line nonsensory cells don’t seem to have high expression of many lateral-line specific genes (eg. klf17). It is likely that these ear-specific genes are, in part, driving the segregation of supporting cell clusters.

Page 10, end of the first paragraph: given all the regeneration data and analysis of notch mutants, it seems like a pretty safe bet that HCs and SCs arise from a common progenitor, even without lineage tracing data.

We have re-written this section (lines 199-205) to better clarify our intent was to identify progenitor cells to root our pseudotime analysis rather than to make any conclusions about progenitor function.

Figure 3: the data set seemed to be highly skewed towards hair cells with relatively few non-sensory cells collected. Is this the case, or are there more hair cells than non-sensory cells?

The feature plot for *atoh1a* is included in Figure 3—figure supplement 2. We include *dla* here as we use it as a marker in later analyses.

Figure 3D: it would be helpful to include a feature plot for atoh1a as dla does not intuitively appear to be a nascent hair cell marker given the large number of progenitors that also express this transcript.

We do find our cell collection of adult cells is skewed towards hair cells, potentially due to unidentified technical issues. We now include a discussion of these caveats, particularly in regards to identifying supporting cell types, in the Discussion section (lines 378-381).

Figure 4B: so are the crista HCs and crista SCs not included in the trajectory?

The reviewer is correct that the crista HCs and SCs were not included in the trajectory in Figure 4B. We have now added a new analysis in Figure 4—figure supplement 1 that shows the trajectory for the crista cells.

Reviewer #3 (Recommendations for the authors):Review of the Results and FiguresThe authors' main goal in the manuscript is to understand the diversity in hair cell and support cell subtypes in the zebrafish inner ear and their relationship to the mammalian inner ear. Overall, the manuscript's readability can be revised to improve the flow of the text, but there is sufficient information presented for readers to follow the rationale, procedures, and insights. The results are clear, and the methods are appropriate.Some recommendations to strengthen the readability and conclusions of the manuscript:The authors find that hair cells and support cells in the inner ear are transcriptionally distinct from the lateral line systems.Figure 1 should include a schematic of the lateral line systems. It should also be described in the figure legend that the zebrafish and mouse inner ears represent adult structures.

We thank the reviewer for the suggestion. A schematic of the lateral line system in a 5 dpf zebrafish has been added to Figure 1 and the figure legend has been updated to clarify the ages represented.

There are 2 paragraphs in the Results section that describe inner ear vs lateral line results on page 7 (of the merged file) and page 11. The results related to the distinct molecular signatures of the lateral line and inner ear should be moved earlier in the text and combined with the results on page 7 to improve the flow of the text. Figure S5 is a great figure showing that the lateral line and inner ear clusters do not overlap which indicates their differences, but this figure is redundant since a similar UMAP is shown in Figure 2B. If the Results section regarding the differences between the lateral line and inner ear are combined, then the authors can also combine Figure S5B with Figure 2. Tables of marker gene data for each partition related to Figure 2B and Figures 2C-F should be provided as a supplement and indicated in the Results section of the text.

We had considered combining the two sections with lateral line and inner ear comparisons. However, the analysis in Figure 3—figure supplement 1 was performed with the 12 mpf cells introduced in Figure 3. We decided that placing the analysis before the introduction of the 12 mpf data would be confusing to readers.

We have computed differentially expressed genes and gene modules for the dataset shown in Figure 2 and added these to Figure 2—figure supplement 1 and Supplemental Table 2.

Figure 2. The colors along the horizontal axes in Figure 2C-2F correspond to the colors in Figure 2B, but they should be clearly described in the figure legend for readers.

We have added clarifying text to the figure legend.

The authors identify distinct hair cells and supporting cell types in juvenile and adult inner ears of zebrafish. They also identify an additional putative progenitor cluster.Figure S2 represents UMAPs of 12 mpf zebrafish inner ear. However, this is the first and only time we are seeing these UMAPs of 12 mpf ears exclusively. It would be beneficial to the reader if there was a UMAP of the unsupervised clustering adjacent to the feature plots. Labels should be included to guide the reader to the structural cells that the author is describing. OR, the authors can remove this figure as it doesn't contribute new information and appears to be a processing step that can be described in the methods section.

We have added an unsupervised clustering UMAP plot to this figure (now Figure 3—figure supplement 1). We believe these data should be included as it indicates how we sub-clustered cells for further analysis.

Figure S3 represents feature plots of genes expressed in the integrated scRNA-seq dataset and clearly displays hair cell genes and support cell genes. The progenitor cell markers selected for this figure do not clearly show that the transcriptional signature of the putative progenitor cell type is distinct from the other cell types. From Figure S3, it appears that the pan-supporting markers overlap with putative progenitor markers. Are there other markers in the progenitor pool from Table S2 that can be selected to display their transcriptional distinctiveness? The authors should draw an outline or point to where the progenitor cell types are in the UMAP plots.

We believe we now more clearly indicate where progenitors are located in plots. While we agree that the progenitor cluster (cluster 0) is not easily distinguished from supporting cell clusters (clusters 6 and 7) in the UMAP plot, they were revealed as distinct by unsupervised clustering (and perhaps due to collapsing from high-dimensional space to two dimensions). Given their relative proximity it is perhaps not surprising that many genes are shared across these groups. Indeed one of the purposes of the pseudotime analysis was to potentially distinguish relative levels of gene expression as defining these groups. We have now included a new supplemental figure (Figure 3—figure supplement 3) to plot the gene expression of putative progenitor markers separately in clusters 0, 6, and 7. Notably, several putative progenitor genes (*pard3bb* and *fat1a*) still retain some expression in supporting cell populations. This is also seen with our new gene module analysis (Figure 3—figure supplement 4) where progenitor signature module genes (Module 1) retain some expression in supporting cell clusters, and supporting cell signature module genes (Modules 8 and 9) can also be found expressed in progenitors.

Figure 3 also displays feature plots of genes expressed in hair cells, support cells, and progenitor cell types. The authors state that the putative progenitor cells show weak expression of hair cells and support cell markers. However, according to Figure 3D, macula and cristae support cells have high expression of the progenitor cell gene Fgfr2. Visually, the observation they write about isn't clear. It might be helpful to include the fold change in gene expression relative to the other clusters to support their observation that the novel progenitor cell genes are enriched.

We have deleted the statement that “putative progenitor cells show weak expression of hair cells and support cell markers". However, we do now note that there is some overlap between progenitor cells and supporting cells in the UMAP space. As shown in new Figure 3—figure supplement 3, some progenitor markers (*Fgfr2*, *igsf3*) are much lower in hair and supporting cells, while other progenitor markers (*fat1a*, *pard3bb*) are reduced but not absent in supporting cells.

The authors propose that the progenitor cells represent a transition state cell type that may contribute to either hair cells or support cells which is also supported by recent work on regenerating zebrafish inner ears. Although lineage tracing is required to validate this idea, the authors explore cell fate trajectories using Monocle3 to examine gene expression as differentiation progresses. They provide the Morans I test results in Table S3, but it would be more informative if the authors clustered genes into modules that are co-expressed across cells. An additional layer of Monocle3 analysis (i.e. gene_module_df) will uncover novel regulatory interactions governing the support cell to progenitor cell transition and the progenitor cell to hair cell transition.

We thank the reviewer for this helpful suggestion as it has allowed us to better separate putative progenitor cells from mature support cells. We now include this gene module information for the embryonic and integrated datasets in Figure 2—figure supplement 1, Figure 3—figure supplement 4, and Supplemental Tables 2 and 5.

Considering that the authors have beautifully labelled hair cell and support cell subtypes to validate their transcriptomic findings, can they also label progenitor cells at the single cell level with spatial information in the inner ear tissues using in situ hybridization assays?

We attempted HCR-FISH with probes for *Fgfr2*, but unfortunately we were not able to distinguish gene expression differences. Instead we performed ISH with the Notch ligand *dla,* which is enriched in in progenitors and is also expressed in nascent hair cells (see dotplots in Figure 3E). HCR-FISH on 5 dpf fish shows *dla* expression in hair cells and in a subset of supporting cells adjacent to hair cells. We have added images of the *dla* ISH in new Figure 4—figure supplement 2. Unfortunately, clarifying progenitor identity and location is going to require further transcriptome analysis, labeling, and lineage tracing, which is beyond the scope of this study.

The authors carefully describe support cell types in cristae versus maculae. According to the methods section, the authors performed single-cell experiments on the lagena. What are the differences between the lagena, saccule, and utricle? Can information about the lagena be extracted from the integrated analysis?

Since we had combined the sensory patches prior to library preparation, we were not able to pull out cells from individual macula organs based on barcode information. Our analysis suggests that, while macula and crista hair cells can be fairly easily distinguished from each other, hair cells from within different macular organs are more difficult to discern. While we were able to identify some markers specific to the utricle *(skor2*+) and saccule (*loxhd1b*+), we find that overall the expression profiles of macular hair cells are relatively similar and hence we were not able to confidently assign an organ of origin. This finding agrees with recent evidence from Jimenez et al., 2022 showing that adult zebrafish utricle and saccule have similar expression profiles.

For Figure 5, it would be helpful to have a schematic of the zebrafish 5dpf inner ear anatomy with labels. Alternatively, a schematic of the 5dpf inner ear can be added to Figure 1.

We have added a schematic of the 5 dpf ear to Figure 1.

Similarly, Figure 6 can include schematics of the adult saccule and utricle OR these structures could be shown in Figure 2. In Figure 6, the utricle, saccule, and lagena should be outlined and have arrows to guide the reader. Figure 8 should also include outlines of the utricles.

We have added schematics of *cabp1b* and *cabp2b* expression patterns to Figure 6. We have also added the outlines of the adult utricle to Figure 8.

Review of the Methods SectionFor Single-cell preparation and analysis, it should be clear whether separate libraries for lagena, saccule, and utricle were prepared or if the dissected structures were combined before library preparation.

We have added a description in Materials and methods (lines 483-484) indicating a single library was built with the dissected structures combined before library preparation.